# Deep-water circulation changes lead North Atlantic climate during deglaciation

Francesco Muschitiello[1,2,3], William J. D'Andrea[2], Andreas Schmittner[4], Timothy J. Heaton[5], Nicholas L. Balascio[6], Nicole deRoberts[2], Marc W. Caffee [7,8], Thomas E. Woodruff[7], Kees C. Welten[9], Luke C. Skinner[10], Margit H. Simon[3] & Trond M. Dokken[3]

Constraining the response time of the climate system to changes in North Atlantic Deep Water (NADW) formation is fundamental to improving climate and Atlantic Meridional Overturning Circulation predictability. Here we report a new synchronization of terrestrial, marine, and ice-core records, which allows the first quantitative determination of the response time of North Atlantic climate to changes in high-latitude NADW formation rate during the last deglaciation. Using a continuous record of deep water ventilation from the Nordic Seas, we identify a ~400-year lead of changes in high-latitude NADW formation ahead of abrupt climate changes recorded in Greenland ice cores at the onset and end of the Younger Dryas stadial, which likely occurred in response to gradual changes in temperature- and wind-driven freshwater transport. We suggest that variations in Nordic Seas deep-water circulation are precursors to abrupt climate changes and that future model studies should address this phasing.

---

[1] Department of Geography, University of Cambridge, Cambridge CB2 3EN, UK. [2] Lamont-Doherty Earth Observatory, Columbia University, Palisades, NY 10964, USA. [3] NORCE Norwegian Research Centre and Bjerknes Centre for Climate Research, 5007 Bergen, Norway. [4] College of Earth, Ocean, and Atmospheric Sciences, Oregon State University, Corvallis, OR 97331-5503, USA. [5] School of Mathematics and Statistics, University of Sheffield, Sheffield S3 7RH, UK. [6] Department of Geology, College of William and Mary, Williamsburg, VA 23187, USA. [7] Department of Physics and Astronomy, Purdue University, West Lafayette, IN 47907, USA. [8] Department of Earth, Atmospheric, and Planetary Sciences, Purdue University, West Lafayette, IN 47907, USA. [9] Space Sciences Laboratory, University of California, Berkeley, CA 94720, USA. [10] Godwin Laboratory for Palaeoclimate Research, Department of Earth Sciences, University of Cambridge, Cambridge CB2 3EQ, UK. Correspondence and requests for materials should be addressed to F.M. (email: francesco.muschitiello@geog.cam.ac.uk)

Precise reconstructions that resolve the relative timing of changes in North Atlantic Ocean circulation, climate, and carbon cycling are necessary to anticipate the mechanisms initiating and propagating abrupt global climate changes. During the last deglaciation (~18,000-11,000-years ago), the climate system underwent numerous abrupt changes that have been attributed to variations in the strength of the Atlantic Meridional Overturning Circulation (AMOC)[1,2]. Through changes in high-latitude North Atlantic Deep Water (NADW) formation and export, AMOC exerts an important control on the global climate system by redistributing heat near the surface and regulating carbon storage at depth. In particular, the partitioning of carbon between the surface and deep ocean is thought to play a critical role in centennial-to-millennial-scale variations of atmospheric $CO_2$ (refs [3–5]). However, reconciling the deglacial history of changes in overturning circulation as recorded in marine records with North Atlantic climate and $pCO_2$ as inferred by Greenlandic and Antarctic ice cores, respectively, remains challenging. First, highly resolved records from deep convection sites sensitive to NADW that monitor the descending branch of AMOC are still lacking. Secondly, large uncertainties in high-latitude marine reservoir ages[6] limit the precision of marine [14]C-based chronologies. Thirdly, direct alignment of marine records to far afield Greenland ice-core stratigraphies hinders testing hypotheses of synchronicity. Lastly, precise comparisons between marine and ice-core climate records are hampered by inconsistencies between the radiocarbon and ice-core timescales[7,8].

Here we present a new synchronization of high-latitude NADW, climate, and $pCO_2$ records for the last deglaciation based on new marine and ice core data that allows us to conclude for the first time that changes in deep-water circulation in the Nordic Seas led rapid shifts in North Atlantic climate and changes in carbon cycling.

## Results

**Site location, [14]C ventilation and chronology**. We generated a continuous record of deep/intermediate- and surface-water [14]C ventilation age from [14]C measurements on planktic and benthic foraminifera (Methods) in sediment core MD99-2284 (62° 22.48 N, 0° 58.81 W, 1500 m water depth) from the Norwegian Sea (Fig. 1).

Site MD99-2284, which is characterised by exceptionally high sedimentation rates (>400 cm kyr$^{-1}$), is located at the gateway of the Faroe-Shetland Channel (FSC), where warm surface Atlantic water flows into the Nordic Seas and cold dense water overflows into the North Atlantic. Critically, this overflow water is one of two main NADW pathways flowing into the deep North Atlantic and a key constituent of the AMOC[9]. During the last glacial period and deglaciation, overflow through the FSC remained a continuous source of NADW[10,11]. Hence, because deep-water [14]C activity reflects the circulation-driven exchange of carbon between the atmosphere and deep-ocean reservoir, bottom-surface water [14]C age differences of the FSC directly inform past changes in Nordic Seas deep convection, NADW production, and its southward export[12,13].

The age model for the core was established using a combination of tephrochronology and alignment between sea-surface temperature records from core MD99-2284 and a high-resolution hydroclimate reconstruction from a relatively closely located terrestrial sequence in southern Scandinavia (Methods; Supplementary Figs. 1–5 and Note 1–2). The approach enables us to precisely place our marine proxies on the IntCal13 timescale[14] and to use the foraminiferal radiocarbon data (Methods) to calculate the marine [14]C ventilation age. Our estimate was determined using a random walk model (RWM) (Methods and Supplementary Methods) fitted via Markov chain Monte Carlo

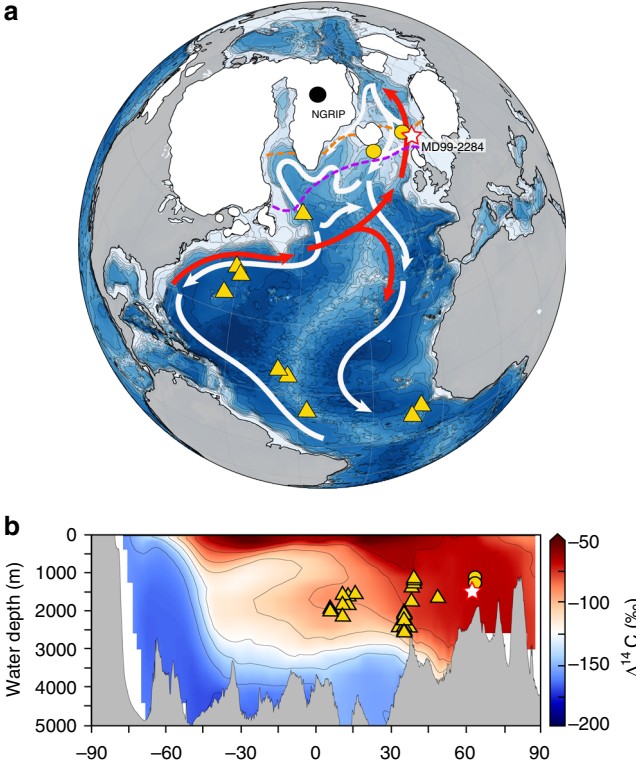

**Fig. 1** Site location. **a** Location of core MD99-2284 (star), NGRIP ice cores (black circle)[101], and other sediment cores (yellow circles) and deep-sea corals (yellow triangles) discussed in this study. The white areas indicate the extent of Northern Hemisphere ice sheets at 12500 years BP[102]. Red and white arrows show warm Atlantic inflow to the Nordic Seas and main bottom current pathways in the northern North Atlantic. Simulated winter (March, purple) and summer (September, orange) 50% sea-ice cover fraction during GS-1 (12,500 years BP)[103] is also shown. **b** Meridional section of radiocarbon concentration within the Atlantic Ocean (averaged over 0–40°W) from a pre-industrial control simulation using a coupled climate-biogeochemical model[104]

(MCMC) that took into account uncertainty structures in both calendar age modelling and [14]C measurements. To allow detailed comparison with Greenlandic and Antarctic ice-core records, we synchronized the ice-core GICC05 (ref. [15]) and WD2014 (ref. [16]), and [14]C timescales using previously published and new [10]Be records from GRIP[17] and WAIS Divide ice cores, respectively (Methods; Supplementary Fig. 6). Ages are hereafter reported as IntCal13 years before 1950 AD ± 1σ (BP).

**Deglacial ventilation history of the deep Nordic Seas**. Surface and bottom water mass reconstructions at our site are consistent with existing paleoceanographic records of water properties, transport and exchange between the Norwegian Sea and the northern North Atlantic (Supplementary Figs. 7–8), indicating that our reconstructions are representative of regional oceanographic conditions. Benthic-planktic (B-P) ventilation ages in MD99-2284 decreased by ~700 years seemingly shortly preceding the abrupt warming transition from Greenland Stadial (GS) 2 (equivalent to Heinrich Stadial 1, HS-1, and the Last Glacial Maximum) into Greenland Interstadial (GI) 1 (equivalent to the Bølling-Allerød interstadial, BA; 14581 ± 16 years BP) (Fig. 2a–d; Supplementary Fig. 9). Although relatively older deep/intermediate water during GS-2 (HS-1) at our site is inferred from only one B-P estimate, this is confirmed by other regional B-P

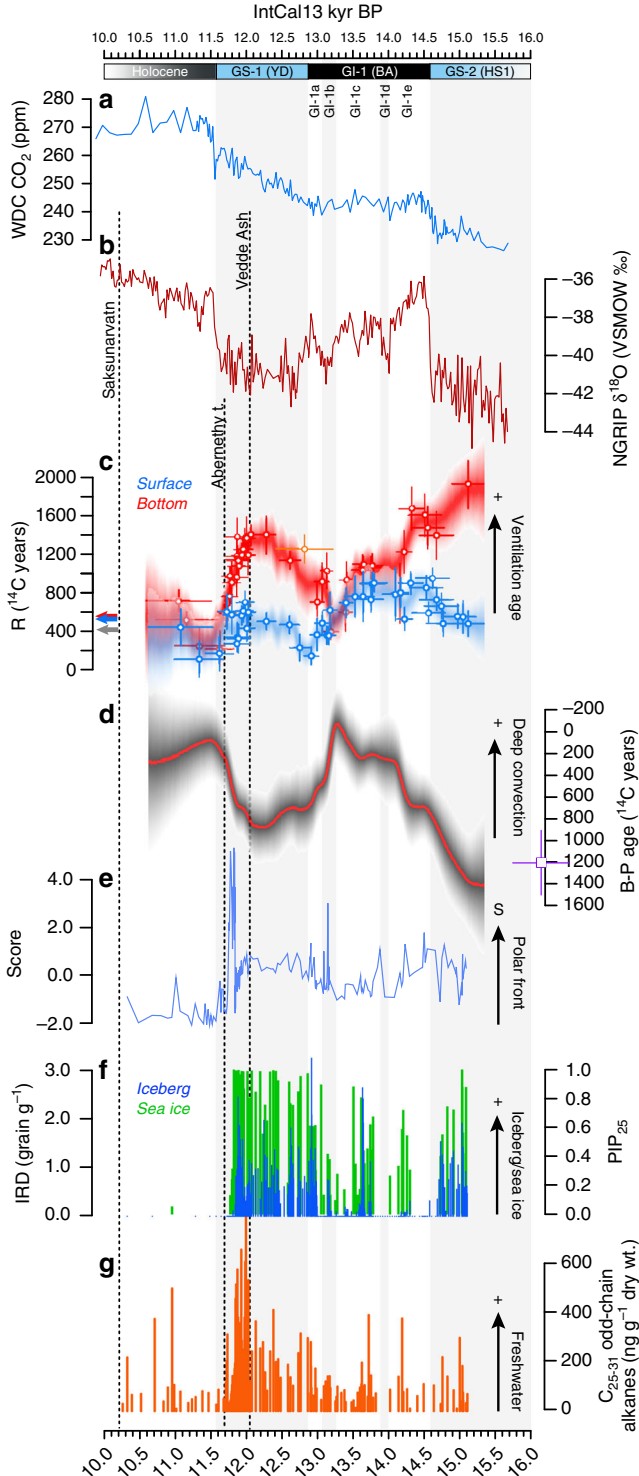

**Fig. 2** Synchronized Antarctic and North Atlantic climate records on the IntCal13 timescale. **a** $CO_2$ concentration from WAIS Divide ice cores (WDC)[3]. **b** $\delta^{18}O$ values from NGRIP ice cores[101]. **c** Bottom and surface $^{14}C$ ventilation (R) histories from core MD99-2284 based on benthic and planktic foraminifera, respectively. Dots indicate individual measurements together with their 2σ error bars. Shading reflects the 95% posterior credible interval (2σ) of $^{14}C$ ventilation as a function of age found by MCMC using random walk model (Methods) and taking into account both the analytical and chronological uncertainty in our observed data. Orange dot reflects individual early GS-1 measurement from nearby core JM11-FI-19PC (1179 m)[12] (not incorporated in the random walk model). Red and blue arrows indicate the modern surface (0–100 m) and bottom (1200–1500 m) R values at the study site[104], respectively. Grey arrow indicates modern mean marine R. **d** Benthic-planktic (B-P) offset based on ventilation estimates in (**c**) and reflecting the strength of deep convection and NADW formation in the Nordic Seas. Red line and shading denote the posterior median value and pointwise 95% credible intervals. Purple square indicates a B-P estimate from the Vøring Plateau (1048 m) based on one solitary U/Th dated deep-sea coral[105] (average of three B-Atm measurements—Supplementary Fig. 9) and presented on its independent time scale. **e** Second principal component of all foraminifera counts in core MD99-2284 dominated by *N. labradoricum*, which is an indicator for the proximity of the Polar Front and sea-ice edge[106] (Supplementary Fig. 10). **f** Coarse-grained (>150 μm) ice-rafted debris count and $PIP_{25}$ index in core MD99-2284 reflecting iceberg rafting and occurrence of spring sea-ice cover, respectively. **g** Abundance of long-chain *n*-alkanes in core MD99-2284 derived from terrestrial higher plants[107] and indicating meltwater discharge. All records are presented on the IntCal13 timescale[14]. Greenland stratigraphic events relative to the IntCal13 timescale are displayed at the top and cold events are highlighted with grey bars (GS: Greenland Stadial; GI: Greenland Interstadial; YD: Younger Dryas Stadial; BA: Bølling-Allerød Interstadial; HS1: Heinrich Stadial 1). Dashed vertical lines show tephra horizons identified in core MD99-2284

relative to convection sites south of Greenland[18]. The century-scale northward order of NADW re-initiation is consistent with other model results[19]. It is also in line with paleoceanographic reconstructions suggesting a delayed resumption of deep convection in the high-latitude Nordic Seas starting midway through GI-1 (BA)[20].

Prior to the onset of GS-1 (12870 ± 26 years BP), which corresponds to the Younger Dryas stadial (YD), the B-P age offset rapidly increased by ∼900 years starting at 13,250 years BP. Ahead of the termination of GS-1 (11577 ± 16 years BP), the B-P offset returned to modern values beginning at 11,890 years BP (Figs. 2d–3). GS-1 (YD) onset and termination are also preceded by surface oceanographic changes inferred from sedimentological and lipid biomarker signatures in MD99-2284. Specifically, increasing (decreasing) B-P ages are virtually synchronous with a southward (northward) migration of the polar front across the coring site inferred from faunal assemblages (Fig. 2e; Supplementary Fig. 10), relatively more (less) frequent sea-ice edge conditions inferred from $PIP_{25}$ (Methods) and iceberg rafting inferred from IRD (Fig. 2f), and relatively higher (lower) inputs of terrestrial meltwater inferred from sedimentary *n*-alkane concentrations (Methods; Fig. 2g), ultimately pointing at a tight link between freshwater dynamics and bottom-water ventilation. Although the benthic-atmosphere (B-Atm) offset dominates the B-P signal, the P-Atm offset also decreased prior to the onset of GS-1 (YD) and in phase with the rise in B-P values (Fig. 2c, d). Specifically, P-Atm values slowly declined by ∼500 years starting ∼13,250 years BP and approached the age of the contemporaneous atmosphere by ∼12900 years BP, suggesting that the climatic transition into GS-1 (YD) was preceded by a major slowdown of the Atlantic Inflow

records (Supplementary Fig. 9). B-P ages gradually decreased during GI-1 (BA) by an additional ∼700 years, reaching or surpassing present-day values (0–50 years) as late as 14,150 years BP (Fig. 2d). This trend suggests that strengthening of deep convection across the onset of GI-1 (BA) was slower than previously thought, which can be attributed to a gradual northward re-initiation of NADW production in the North Atlantic. This interpretation is supported by transient simulations of the last deglaciation, which reveal a northward time-transgressive recovery of NADW formation during GI-1 (BA), with a ∼400-year long re-initiation of deep convection in the Nordic Seas

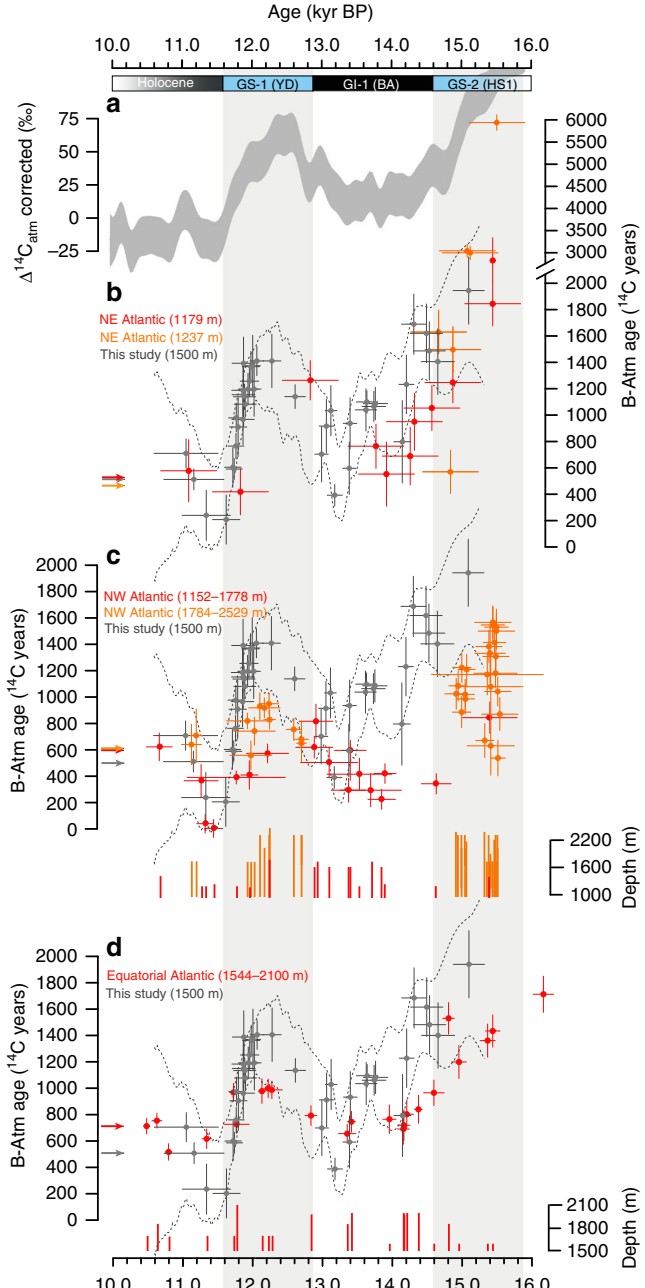

**Fig. 3** North Atlantic deep/intermediate water $^{14}$C ventilation during deglaciation. **a** High-pass filtered (cut-off frequency 1/200 years) atmospheric radiocarbon content ($\Delta^{14}$C) corrected for changes in natural $^{14}$C production[27], reflecting changes in global ocean ventilation and NADW formation[28,29]. **b** Comparison of bottom $^{14}$C ventilation histories (B-Atm) from core MD99-2284 (grey dots and dashed lines) and B-Atm records from the Faroe-Shetland Channel (1179 m)[12] (red) and south of the Iceland-Scotland Ridge (1237 m)[12] (orange) using benthic foraminifera (excluding *Pyrgo spp.* and Miliolids) (note the broken *y*-axis and scale change at ~2500 $^{14}$C years). **c** Same as in (**b**) using intermediate coral $^{14}$C records from Orphan Knoll[108] and the New England Seamounts[22–25] and deep/intermediate coral $^{14}$C records from the New England Seamounts[22–25] in the northwest Atlantic Ocean (Fig. 1). **d** Same as in (**b**) using deep/intermediate coral $^{14}$C records from the Equatorial Atlantic[26]. Arrows show modern depth-averaged ventilation ages at each location. Histograms display depth represented by individual data points. Note that the $^{14}$C record from ref. [24] used here includes one measurement from the northeast Atlantic. Bars reflect 2σ errors of individual measurements. All records are presented on their independent time scale. Greenland stratigraphic events relative to the IntCal13 timescale are displayed at the top (GS: Greenland Stadial; GI: Greenland Interstadial; YD: Younger Dryas Stadial; BA: Bølling-Allerød Interstadial; HS1: Heinrich Stadial 1)

of North Atlantic deep/intermediate water $^{14}$C signatures associated with high-latitude NADW ventilation rates. In addition, the timing and pattern of the reconstructed NADW changes are consistent with shifts in production-corrected atmospheric $\Delta^{14}$C (ref. [27]) (Figs. 3, 4), which primarily reflects changes in ocean ventilation and high-latitude NADW formation[28,29], and thus support an oceanic origin for the variations in atmospheric radiocarbon concentrations during the last deglaciation. Overall, the agreement across the range of independent deep ventilation records supports the view that our B-Atm data capture large-scale fluctuations in NADW circulation. A strong coupling between AMOC and deep ventilation in the Nordic Seas is also corroborated by transient climate model experiments (Supplementary Fig. 11 and Note 3). The simulations, which reproduce a shutdown and subsequent recovery of the AMOC, identify and validate sign and magnitude of the shifts observed in our B-P age reconstruction.

A detailed comparison between our B-P record and ice-core-based temperature records from Greenland using breakpoint analysis (Methods; Supplementary Table 1; Fig. 4), reveals a complex relationship between Nordic Seas ventilation and abrupt climate change. The analysis reveals that changes in Nordic Seas NADW formation occurred before the climate shifts into and out of GS-1 (YD) by 385 ± 32 (1σ bounds) and 447 ± 27 years, respectively, and that weakening of Nordic Seas NADW occurred 437 ± 79 years prior to the first signs of $p$CO$_2$ rise near the start of GS-1 (YD). Importantly, the latter finding substantiates the hypothesised[30,31] lag between AMOC reduction and $p$CO$_2$ rise during early deglaciation and is observed in climate simulations as a transient response of the global efficiency of the biological pump to AMOC slowdown (Supplementary Figs. 12–14 and Note 3).

The lag between our high-latitude NADW records and Greenland temperatures across the transitions into and out of GS-1 (YD) requires further consideration. Cooling of sea-surface temperatures and sea-ice expansion in the Norwegian Sea preceding GS-1 (YD) have been documented in terrestrial temperature and isotope records[32,33], and are consistent with increasing B-P offsets observed in MD99-2284 beginning at 13,250 years BP. Beyond the last deglaciation, records from the last glacial and Late Pleistocene[34,35] have also implied increased

that resulted in enhanced isotopic equilibration between surface waters and the atmosphere[21].

## Discussion

The timing and magnitude of changes in our B-Atm record are supported by other independent reconstructions from the FSC and the Iceland Basin, i.e., regions that monitor ventilation of Nordic Seas overflow waters[12] (Fig. 3). The data are also in agreement with deep ventilation records from U–Th dated corals in the deep/intermediate northwest and equatorial Atlantic, which monitor the strength of downstream NADW transport[22–26]. Specifically, the slow decrease in B-Atm ages at the end of GS-2 (HS-1) and during the early GI-1 (BA) is observed in data from both the northeast and equatorial Atlantic, whereas early changes prior to the start and end of GS-1 (YD) are evident in each of the records. We hypothesise that these parallel changes in radiocarbon across sites likely reflect downstream propagation

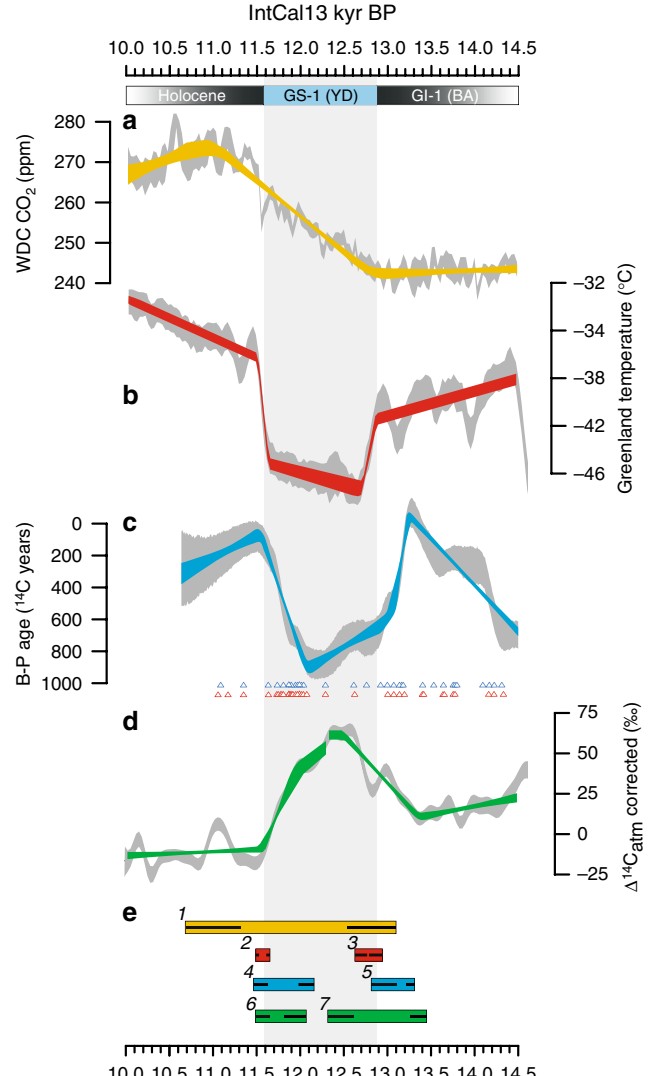

**IntCal13 kyr BP**

**Fig. 4** Detailed view of synchronized $CO_2$, climate and ocean circulation records during the Younger Dryas stadial. **a** $CO_2$ concentration from WDC ice cores. **b** Greenland temperature reconstruction[109] (average of NEEM, GISP2 and NGRIP ice cores). **c** Nordic Sea deep water formation rate based on B-P ages from core MD99-2284 (this study). Triangles indicate the location of individual planktic (blue) and (benthic) $^{14}C$ measurements. **d** Atmospheric radiocarbon content ($\Delta^{14}C$) corrected for changes in natural $^{14}C$ production[27]. The record was high-pass filtered (cut-off frequency 1/200 years) to facilitate comparison with long-term overturning strength variations (**c**). All the records were generated by MCMC using the same random walk model approach used to estimate $^{14}C$ ventilation, but the model was fitted directly to the observations. Grey envelopes reflect the 68% credible intervals (1σ) associated with both analytical and chronological uncertainty in the raw data. Coloured segments represent the 68% confidence intervals of the fitted piecewise linear regression functions accounting for both analytical and chronological uncertainties. The regression models were estimated using a modified version of an algorithm for breakpoint analysis[100]. Note that only the sharp decline in $\Delta^{14}C$ during the second half of GS-1 (YD) can be attributed to a resumption of NADW formation[27], whereas the preceding minor decline is likely a dynamical response in the Southern Ocean associated with venting of $^{14}C$-depleted carbon[110]. **e** 95% confidence intervals of the timing and duration of the transitions (Supplementary Table 1). Inset black lines show the 2σ uncertainty of the estimated start and end of each transition. Greenland stratigraphic events relative to the IntCal13 timescale are displayed at the top (GS: Greenland Stadial; GI: Greenland Interstadial; YD: Younger Dryas Stadial; BA: Bølling-Allerød Interstadial)

input of ice-sheet meltwater, gradual surface cooling, and southward migration of the polar front in the Nordic Seas pre-ceding transitions from warm interstadials to cold stadial con-ditions, analogous to the oceanographic changes inferred from MD99-2284 that preceded GS-1 (YD).

On the other hand, ocean warming in the Nordic Seas, a northward diversion of the polar front, and NADW resumption during the second half of the GS-1 (YD) have been widely reported[36–38], and further support decreasing B-P offsets in MD99-2284 starting at 11,890 years BP. A lead of a few centuries of changes in high-latitude NADW formation ahead of Green-land temperatures is compatible with the timing of AMOC slowdown and recovery during deglaciation inferred from several Pa/Th circulation[39] and other AMOC-sensitive proxies[40,41]. Despite uncertainties with the $^{14}C$ chronologies and coarse temporal resolution, it has been demonstrated that a century-scale response of Pa/Th to AMOC changes should be additionally accounted for when interpreting this proxy as a function of changes in ocean circulation[42,43]. The existence of a significant lag time for Greenland temperature and Pa/Th records behind changes in NADW would represent an important observational constraint to account for, for example when attempting to infer the physical mechanisms of rapid climate and AMOC change from climate model simulations[44,45]. Equally abrupt and synchronous changes in the AMOC, Pa/Th and Greenland

temperature (i.e., to within a few centuries) should not, therefore, be expected in numerical model simulations.

On the basis of timing inferred from our $^{14}C$ ventilation records, we propose that the lagged Greenland temperature response behind changes in Nordic Seas NADW formation can be understood as a threshold response to gradual changes in surface temperature-driven freshwater transport, which ulti-mately modulates the strength of the northward oceanic heat transport. Both empirical[34,35,46] and climate modelling studies[47] have proposed a nonlinear salt oscillator in the North Atlantic system, whereby meltwater production rates are greater (smaller) toward the end of warm (cold) episodes. This is consistent with evidence for southward (northward) migration of the polar front, more (less) extensive sea-ice cover, and increased (decreased) iceberg discharge at our coring site shortly before the termination of GI-1 (GS-1). We further hypothesise that shifts to higher (lower) rates of meltwater fluxes, although initially prompted by gradual climate warming (cooling), are amplified by concomitant changes in surface winds, which play a critical role in the rear-rangement of water masses in the Nordic Atlantic. For instance, model studies suggest that sustained cold events like GS-1 (YD)[48] and the Little Ice Age[49,50] can be triggered by a positive feedback loop, whereby increasing export of meltwater to the subpolar gyre in response to warming leads to surface ocean cooling and con-sequent strengthening of atmospheric blocking over the North Atlantic–i.e. large-scale quasi-stationary anticyclonic circulation. This feedback can impede northward heat transport and suppress NADW formation for centuries. Moreover, reduced NADW can lead to an additional freshening through basin-wide subsurface warming[51,52], which causes ice-shelf thinning and iceberg dis-charge[52], further bolstering NADW formation weakening. However, we cannot rule out the potential amplifying role of a catastrophic drainage of freshwater from the Baltic Ice Lake into the Nordic Seas[53], which occurred precisely at the start of GS-1 (YD), and could have further impacted regional NADW formation.

Atmospheric blocking could be equally important as a mechanism for NADW resumption in the Nordic Seas at the transition out of GS-1 (YD). New studies suggest that under full GS-1 (YD) (ref. [54]) and glacial conditions[55], the cold Fennoscandian Ice Sheet induces a strong southwesterly wind flow over the Norwegian Sea associated with blocking circulation over the ice sheet. This configuration is more conducive to reduced sea-ice cover through export of sea ice out of the Norwegian Sea and increased surface warming and salinity in response to northward oceanic transport of heat and salt, which in turn gradually promotes deep-water formation in the Norwegian Sea[55]. Our data and other existing records all lend support to the occurrence of this proposed mechanism midway through GS-1 (YD) (i.e., stronger southwesterly winds[36,37], warmer surface ocean conditions[56,57], and gradually less frequent sea-ice occurrence[58,59]) (Supplementary Fig. 7).

In conclusion, our results suggest that gradual changes in high-latitude NADW formation are precursors of rapid climate shifts in the North Atlantic and emphasize the central role of ocean circulation in abrupt climate change, as well as its sensitivity to atmosphere and cryosphere dynamics. Given recent evidence that the current AMOC has been slowing down for several decades[60,61], our findings broach the question as to whether the current decline in deep-water circulation may herald a new phase of abrupt change.

## Methods

**Chronology**. The chronology of core MD99-2284 was established by aligning variations in downcore sea-surface temperatures (SST)[36,62] with synchronous changes in hydroclimate as recorded in sedimentary leaf-wax hydrogen isotope (δD) records from the ancient lake of Atteköp, southern Sweden[63] (Supplementary Figs. 1–5 and Note 1). The terrestrial site is located closely downwind of our marine core location in an area where most of the precipitation is sourced from the North and Norwegian Seas[64], and where hydroclimate shifts have a demonstrated modern and past linear relationship with upwind, near-field upper ocean temperatures via changes in sea-to-air-moisture fluxes (Supplementary Figs. 1–2). Leaf-wax δD records in northern Europe have been successfully used as sensitive indicators of annual changes in the isotopic composition of the marine precipitation source and moisture availability[33,65]. Therefore, precise synchronization of marine SST and terrestrial δD records in this region affords an accurate chronology for core MD99-2284 that circumvents alignment to the far-afield Greenland ice-core isotope stratigraphy, which is not necessarily representative of near-surface paleoceanographic conditions in the North and Norwegian Seas[33].

Synchronization between the marine and terrestrial proxy time series was obtained using an automated stratigraphic alignment algorithm driven by a Markov chain Monte Carlo method[33,66] run for $10^6$ iterations. The approach involves nonlinear deformation of the SST time series onto the reference δD time series to deliver an optimal alignment accounting for uneven compaction and/or expansion of sediments over time, as well as for analytical errors associated with the proxy data (Supplementary Fig. 3). An account of the mathematical formulation associated with the algorithm is presented in ref. [67].

Atteköp's chronology was constructed using a Bayesian age model based on the Hässeldalen Tephra and 37 AMS $^{14}$C dates from selected terrestrial plant macrofossils[63] (Supplementary Fig. 3) calibrated with the IntCal13 curve[14], which in turn allows placing MD99-2284 records on the atmospheric IntCal13 timescale. However, the alignment between the marine and terrestrial records was done only up to ~12,600 years BP due to the scarcity of datable plant macrofossils in Atteköp in this portion of the core (Supplementary Fig. 3). To secure the chronology upcore, we thus employed three well-dated tephra layers identified in MD99-2284 sediments–the Vedde Ash, the Abernethy Tephra, and the Saksunarvatn Ash (Supplementary Note 2)—and fit the tephra-based age constraints using a Gaussian Regression model[68].

**Tephra analysis**. Two important regional isochrones, i.e. the Vedde Ash and the Saksunarvatn Ash (both present in Greenland ice cores[69]), have been previously reported in core MD99-2284 (ref. [36]). These horizons have been precisely and accurately radiocarbon dated in Lake Kråkenes, Norway[70]. The markers are here supplemented by the first reported marine occurrence of the Abernethy Tephra[71,72] (Supplementary Fig. 4). The finding constitutes the outcome of a targeted search for late YD tephra based on the counting of glass shards within the interval 200–360 cm in the marine core. A peak in cryptotephra shards was identified in samples 247.5 cm and 249.5 cm. The shards were isolated from the sediment following the methods of ref. [73]. Samples were treated with 10% $H_2O_2$ and heated to remove organic material, then washed with deionized water over a

63-μm sieve. A series of heavy liquid density separations using sodium polytungstate were then performed to isolate material between 2.2 and 2.5 g cm$^{-3}$, which was mounted on 27 × 46 mm glass slides in epoxy resin. Slides were polished to expose grain interiors and analyzed at the Concord University Microanalytical Laboratory (WV, USA) using an ARL SEMQ electron microprobe equipped with six wavelength-dispersive spectrometers and a Bruker 5030 SDD energy-dispersive spectrometer. Instrument conditions included a 14 kV accelerating voltage, 10 nA beam current, and beam size of 4 to 6 μm. Results were reported as non-normalized major oxide concentrations (Supplementary Data 1).

Results from the analysis of tephra grains in samples 247.5 cm and 249.5 cm show a single geochemical population of rhyolitic composition similar to the widespread Vedde Ash erupted from the Katla volcano in Iceland during the middle of the YD. Recently, it has been shown that there was a second tephra-producing eruption of Katla within the YD, but closer to the YD-Holocene boundary named the Abernethy Tephra[72]. Tephra layers from Loch Etteridge (LET-5)[72] and Abernethy Forest (AF555)[71], in Scotland, were attributed to this eruption, which are both similar in stratigraphy and geochemistry to tephra that we have identified (Supplementary Fig. 4). Based on precise varve counting[72] and other lines of evidence[74,75], it has been argued[75] that the Abernethy Tephra is likely 320 ± 20 years younger than the Vedde Ash (12,064 ± 48 years BP)[70]. Therefore we here assigned to the layer an age of 11,744 ± 50 years BP. Radiocarbon dating of this layer based on one terrestrial plant macrofossil[71] from a sediment sequence in Abernethy Forest yields a calibrated age range of 12,380–11770 years BP, which is consistent with, or slightly underestimates our age assignment.

**$^{14}$C dating**. Well-preserved monospecific shells of the planktic foraminifer *Neogloboquadrina pachyderma (s)* (calcification depth ~30 to 200 m (ref. [76])), two samples of the benthic foraminifer *Cibicidoides wuellerstorfi*, one shell of *Pyrgo murrhina*, and a number of mixed benthic foraminifera were hand-picked from slices of core MD99-2284 for AMS $^{14}$C dating for a total of 88 measurements (41 planktic and 47 benthic), including 8 previously reported dates[36] (Supplementary Data 2). Mixed benthic samples are mainly composed by *C. neoteretis* excluding *Pyrgo spp* and *miliolid spp*. Samples were mainly selected from the >150 μm fraction, except for 31 samples, which were picked from the 63-150 μm fraction. All shells were cleaned and dried prior to preparation for accelerator mass spectrometry (AMS) measurements, whereas large samples were additionally rinsed with dilute, organic-free HCl.

Large samples were converted to graphite and analysed using standard AMS $^{14}$C measurement procedures within the laboratories of Beta Analytic Miami (FL), USA, and ETH Zurich, Switzerland. All the small fraction samples were processed and analysed using a newly developed method[77]. The approach involves direct analysis of $CO_2$ from ultra-small amounts of carbonate (~0.5 mg) using a compact AMS facility equipped with a gas ion source at the Laboratory of Ion Beam Physics, ETH Zurich. To test for contamination by secondary carbonates on the small fraction samples, we performed leaching experiments on the sample material surface using HCl 0.02 M and following the procedure outlined in ref. [78]. All radiocarbon ages are here reported according to the standard protocol of ref. [79]. A subset of 7 samples were prepared at the University of Cambridge using methods detailed in ref. [80] and subsequently dated by AMS at the 14 Chrono Centre, University of Belfast.

The resulting $^{14}$C ages served as a basis to generate subsurface (~100 m) and bottom water (1500 m) ventilation records employing the random walk model described below, and ultimately to reconstruct benthic-planktic (B-P) ventilation ages[81]. We identified seven outliers that were dismissed from our bottom water ventilation record (one in GS-1 at 400.5 cm; three in GI-1 at 415, 445.5, 534.5 cm; three in GS-2 at 576.5, 581.5, 601 cm). These were all ages based on benthic foraminifera, including one *Pyrgo spp.* age, which has been shown to yield old $^{14}$C ages[12]. The other benthic dates, which are all based on small shell fragments <1 mg in weight and all associated with the 63-150 μm fraction, were either significantly younger than the corresponding planktic ages (where available), or significantly younger than the contemporaneous atmosphere. Acid leaching tests indicate that four out of the seven outliers are likely affected by modern $CO_2$ carbon contamination, as evidenced by the high $^{14}$C content in the respective leach fractions (Supplementary Data 2). Given that sedimentation rates at our site are remarkably high (>400 cm kyr$^{-1}$), we argue that complications associated with bioturbation and differential mixing effects are likely negligible[82,83]. Rather, we argue that the systematically younger ages measured in these samples may be the result of preferential modern-carbon contamination among the finer carbonate fraction through secondary growth of calcite within the sediment pore waters and exchange between calcite and the carbonate ions in pore waters[83].

**Random walk model**. Regional reservoir (ΔR) and reservoir (B-Atm and P-Atm) estimates were inferred using a random walk model that incorporates both the uncertainty in the calendar age modelling and our $^{14}$C measurements (Supplementary Data 3). B-P estimates were obtained by subtracting the modelled B-Atm and P-Atm curves. This approach was preferred to the projection method, which is more influenced by atmospheric Δ$^{14}$C variations in the North Atlantic[84]. In the following, we provide a brief overview of the random walk model and the construction of the reservoir age curves.

For each of our $N$ ocean objects, we observe an estimate $t_i$ of its true calendar age $\theta_i$; and a radiocarbon determination $z_i$ that is also subject to noise. This

radiocarbon determination can be decomposed into the site-general marine radiocarbon age $m(\theta_i)$; and $\Delta R(\theta_i)$ the local reservoir variation i.e., difference between the reservoir age in the Norwegian Sea. We aim to estimate the Norwegian Sea $\Delta R(\theta_i)$ given these paired observations $(t_i, z_i)_{i=1,\dots,N}$ and our prior knowledge about the value of $m(\theta_i)$ provided by the Marine13 radiocarbon calibration curve (ref. [14]). We then infer the value of the absolute Norwegian Sea reservoir correction $R(\theta)$ by comparison with the atmospheric radiocarbon curve at the same $\theta$. Specifically we model

$$t_i = \theta_i + \epsilon_i$$

$$z_i = m(\theta_i) + \Delta R(\theta_i) + \eta_i, \tag{2}$$

where $\epsilon_i$ and $\eta_i$ are the uncertainties in our measurement of calendar age and radiocarbon age assumed here to be independent and identically distributed (IID) with mean 0 and variances $\tau_i^2$ and $\sigma_i^2$, respectively. Note in particular that the true calendar ages $\theta_i$ are only observed subject to noise. This should be incorporated into any estimation procedure. If we ignore this additional uncertainty we are likely to introduce a bias in our estimate for $\Delta R(\theta_i)$ and oversmooth.

We take a Bayesian approach for our estimation of $\Delta R(\theta_i)$. This allows us to incorporate our prior beliefs about the value of $m(\theta_i)$; model the evolution of $\Delta R(\theta_i)$ over time in a physically interpretable way; and update these prior beliefs in a consistent framework. To make our problem identifiable we require strong prior information on the site-general $m(\theta)$. This is provided by Marine13, the marine radiocarbon calibration curve[14]. At any calendar age $\theta$ this provides pointwise estimates of the value

$$m(\theta) \sim N\left(\mu(\theta), \sigma_c^2(\theta)\right). \tag{3}$$

Additionally, we require a prior model for the evolution of the Norwegian Sea's reservoir age $\Delta R(\theta)$. While we do not expect this to be constant over time, we would expect that knowing $\Delta R(\theta)$ at a particular $\theta$ provides some information about its likely value at nearby times. We, therefore, place a Wiener process (random walk) prior on $\Delta R(\theta)$ similar to that used in the modelling of the IntCal atmospheric calibration curve (see refs. [85–87]) whereby

$$\Delta R(\theta) | \Delta R(\theta^*) \sim N(\Delta R(\theta^*), \rho|\theta - \theta^*|), \tag{4}$$

The above random walk model is then combined with our prior on $m(\theta)$ and our observed $(t_i, z_i)_{i=1,\dots,N}$ through a Metropolis-within-Gibbs sampler. This provides a posterior estimate for $\Delta R(\theta)$ that allows us to borrow strength from the surrounding observations (i.e., neighbouring $\theta$'s). It also updates our site-general marine $m(\theta)$. Combining both the estimate of $\Delta R(\theta)$ and the updated $m(\theta)$ and comparing with the atmospheric radiocarbon levels provided by IntCal13 (ref. [14]) then allows us to estimate the absolute Norwegian Sea reservoir correction $R(\theta)$, i.e., B-Atm and P-Atm. Details on the implementation of this sampler can be found in Supplementary Methods.

**[10]Be measurements**. A total of 170 samples from 1793 to 2279 m depth in the WAIS Divide 06A ice core (WDC-06A) were analysed for [10]Be concentrations at UC Berkeley and at the Purdue University PRIME Lab (Supplementary Data 4). Samples typically represent continuous ice core sections of ~3 m length (although they varied from 1.9 to 3.9 m) corresponding to ~30 years of snow accumulation. Ice samples of 300–700 g were weighed, melted, and acidified with a solution containing ~0.18 mg Be carrier. The samples were passed through 30 micron Millipore filter, and loaded on cation exchange columns from which the Be fraction was eluted following established procedures[88,89]. The [10]Be/[9]Be ratios of samples and blanks were measured by AMS at PRIME Lab. Absolute ratios were obtained by normalizing the measurements to well-documented [10]Be/[9]Be standards[90]. Results were corrected for an average blank [10]Be/[9]Be ratio of 15 ± 3, which corresponds to typical blank corrections of 1–3% of the measured [10]Be/[9]Be ratios. The [10]Be results up to 12,000 years BP were used by ref. [16] for the synchronization of the WDC core with the Intcal13 [14]C record.

**Timescale synchronization**. Solar activity cycles modulate the amount of Earth bound cosmic rays that controls the production rates of cosmogenic radionuclides [10]Be and [14]C in the upper atmosphere. The former are integrated in ice cores; the latter are incorporated in a number of absolutely dated records (e.g., tree rings) that take up carbon directly from the atmosphere at the time of formation. Hence, matching of the globally synchronous common short-term [10]Be and [14]C variations as reconstructed in ice cores and IntCal13 records[14], respectively, provides a unique tool for estimating timescale differences in a nearly continuous fashion[7,8].

Here we aligned relative changes in GRIP [10]Be flux[17] (on the GICC05 timescale[15]) and new WAIS Divide 06A ice core (WDC) [10]Be concentration data (on the WD2014 timescale[16]) to [14]C from IntCal13 using a Bayesian wiggle match approach to estimate the likely time offset between the ice and radiocarbon timescales (Supplementary Data 5). The [10]Be data are compared to the atmospheric [14]C production rate ($p$[14]C) simulated from IntCal13 $\Delta$[14]C data using a Box-Diffusion carbon-cycle model[91] initialized with pre-industrial boundary conditions assuming a constant carbon cycle. Even though this assumption does not necessarily hold for climatically unstable period such as the YD, sensitivity experiments[8] have shown that centennial-scale variations in atmospheric [14]C

(such as those analysed in this study) are largely unaffected by the dynamics of the carbon cycle and predominantly reflect solar cycle modulations. To estimate the uncertainty associated with carbon cycle effects on the simulated $p$[14]C, we performed four additional sensitivity experiments where the carbon cycle was perturbed by increasing or decreasing the air-sea gas exchange and the ocean diffusivity parameter by 50%, individually. These tests resulted in an uncertainty estimate of ± 2%, which was added to the nominal IntCal13 $\Delta$[14]C error.

Prior to wiggle matching, the cosmogenic nuclide records were bandpass filtered between cut-off frequencies of 1/100 and 1/500 years to focus on the common (likely solar-induced) secular changes in production. Low-pass filtering eliminates short-term weather noise and sampling inconsistencies between the data sets. On the other hand, high-pass filtering minimizes systematic errors associated with millennial [10]Be production rate variability and uncertainties accompanying longer-term climate and carbon cycle changes[92]. We also apply a 1-year lag for [10]Be to account for the average delay between production and deposition, and ultimately scaled the [10]Be time series by a factor of 0.7 to optimize the fit with the $p$[14]C data.

Bayesian wiggle matching was performed on the filtered and scaled data following the mathematical formulations of ref. [93] and methodology of ref. [8] (Supplementary Fig. 6). We sequentially offset-minimized the [10]Be data against $p$[14]C with moving time windows of length of 1000 years at 20-year time steps allowing for leads/lags ranging ±150 years applied to the ice core timescales relative to IntCal13. The probabilities of each individual lead-lag measurement were combined to obtain an overall timescale shift likelihood at each time step between GICC05 and IntCal13, and between WD2014 and IntCal13, respectively. The results allowed us to extend the GICC05-IntCal13 and WD2014-IntCal13 timescale transfer functions of ref. [8] and ref. [16], respectively, beyond the Holocene until ~14,500 years BP. The estimated timescale offsets are well within the stated uncertainties of the ice core timescales (Supplementary Fig. 6), while the associated uncertainties estimates are consistent with the previous results[8]. We also note that our timescale transfer functions are in excellent agreement with previously published estimates over the interval of overlap[8,16]. Finally, Greenlandic and Antarctic ice core records presented in this study were consistently placed on the IntCal13 timescale, whereas beyond the limits of the synchronization described above, the cumulative age uncertainties associated with each ice-core timescale were reset. Note that when applying the timescale correction to WDC $CO_2$ data[3], ice-age–gas-age differences[16] were fully accounted for.

**Lipid biomarkers**. One hundred and sixty-nine sediment samples were collected from core MD99-2284 at 1-cm intervals. The samples (~3 cm³) were freeze-dried, homogenized, and lipids were extracted from the sediments via sonication with a 9:1 mixture (v/v) of dichloromethane:methanol for 20 min. The procedure was repeated three times and supernatants were combined. Total lipid extracts (TLE) were evaporated in a Biotage TurboVap under a stream of nitrogen gas. TLE were separated into constituent lipids by flash silica-gel chromatography (pre-cleaned 100% active silica gel) in hexane-saturated columns using as eluting solvents hexane (F1), dichloromethane (F2), 3:1 hexane:ethyl acetate (F3), and methanol (F4), separately.

To monitor changes in terrestrial meltwater discharge and paleo sea-ice conditions at the coring site, we quantified the abundance of long-chain $n$-alkanes and mono-unsaturated $C_{25}$ highly branched isoprenoid alkene (IP$_{25}$)[94], respectively, contained in the F1 fraction of 169 samples (Fig. 2 and Supplementary Fig. 7). The latter biomarker is biosynthesised by Arctic sea ice diatoms during spring and it has been widely used for paleoceanographic reconstructions in this sector of the North Atlantic[58,95]. The F1 fraction was evaporated and transferred to GC vials. Both $n$-alkanes ($C_{13}$–$C_{39}$) and IP$_{25}$ were identified and quantified on an Agilent 7890A gas chromatograph (GC) equipped with a mass selective detector (MSD). Samples were measured with an Agilent HP-5ms column (30 m, 0.25 mm ID, 0.25 µm film thickness). The GC oven was held for 1.5 min at 60 °C and ramped to 300 °C at 10 °C min⁻¹. After 1 min it was ramped to 320 °C at 2 °C min⁻¹ and held for 10 min. We used a PTV injector in splitless mode, with samples injected at 60 °C and the injector temperature immediately ramped to 320 °C at 900 °C min⁻¹. All the compounds were identified in SIM mode based on mass spectra and retention times from the literature (monitoring ions $m/z$ 57.1, 85.1, 99.1, 245.3, 346.3, 348.3, 350.3). A standard mixture of $n$-alkanes ($C_{10}$–$C_{40}$) was run after every sixth sample injection, and was used to evaluate instrument drift and correct for mass-dependent differences in the response factor of the MSD. Concentrations of individual $n$-alkanes were calculated based on the comparison of the adjusted peak areas relative to both $C_{36}$ alkane and 5α-androstane used as an internal standard, and each added to samples before GC analysis. Similarly, IP$_{25}$ was quantified based on comparison to the internal standard of 9-octyl-heptadec-8-ene (9-OHD)[96] added before GC analysis. The measurement accuracy of both $n$-alkanes and IP$_{25}$ was determined by monitoring the relative response factor for $n$-$C_{36}$ and 5α-androstane, and for IP$_{25}$ and 9-OHD (over a range of concentrations), respectively, every seven samples (1σ = ± 3.2%).

To quantify changes in marine surface water phytoplankton productivity, we measured the concentration of $C_{37}$ unsaturated alkenones[97] contained in the F2 fraction of 154 samples (Supplementary Fig. 7). Alkenones were recovered by liquid-phase separation using toluene, and diluted for analysis with a Thermo TRACE Ultra Gas Chromatography Flame Ionization Detector (GC-FID) equipped

with PTV injector operated in splitless mode. Samples were measured with a DB-1 column (60 m, 0.25 μm ID, 0.1 μm film thickness). Lastly, an in-house alkenone standard with known relative concentrations of $C_{37}$ unsaturated alkenones was injected every six samples to monitor the instrument performance and analytical precision of the alkenone-unsaturated index $U^K_{37}$ (ref. [98]) ($1\sigma = \pm 0.0072$).

Changes in sea-ice cover were finally reconstructed by calculating a phytoplankton-IP$_{25}$ index (PIP$_{25}$)[99] (Fig. [2]) following the equation

$$PIP_{25} = [IP_{25}] \frac{[IP_{25}]}{[IP_{25}] + ([P] \times c)} \qquad (5)$$

whereby $[IP_{25}]$ and $[P]$ represent the respective concentrations of IP$_{25}$ and the phytoplankton biomarker P, while $c$ is a balance factor calculated from the ratio of the mean IP$_{25}$ to P concentrations.

**Breakpoint analysis.** To estimate the timing of the transitions recorded in the proxy data, we used a Monte Carlo version of the function Segmented in the R package 'Segmented'[100], i.e., a piecewise linear fitting regression method that determines the breakpoints of two lines. To combine the analytical and age uncertainties associated with each proxy record, while at the same time enforcing monotonicity in the age estimates, the data sets were first modelled using the Random Walk Model described above, whereby the model was fitted directly to the observations. We subsequently randomly resampled (5000 times) the modelled data within its uncertainty, assuming a Gaussian error distribution. We then estimated the breakpoint structure of each Monte Carlo realization using the Segmented function run with 100 bootstrap iterations. The inferred likelihood distribution of each breakpoint allowed us quantifying the timing uncertainty for the start and end of the transitions under investigation (Fig. [4]; Supplementary Table 1).

## Code availability

All the climate model output and R codes used for the numerical procedures are available from the main author upon reasonable request.

## Data availability

The source data underlying Fig. [2]c, d, Supplementary Figures 4 and 6d, f and h (tephra geochemistry, [14]C ages, WDC [10]Be data, ocean-atmosphere [14]C disequilibrium estimates, IntCal13-GICC05 and IntCal13-WD2014 age offsets) are provided as a Source Data file along the online version of this article on the publisher's website and in the PANGAEA paleoclimate data archive.

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

## Acknowledgements

We are grateful to D.I. Blindheim for assistance with foraminifera sampling, H. Sadatzky for assistance with sampling for lipid biomarker analyses, and S. Rasmussen for helpful discussions. The research leading to these results has received funding from the European Research Council under the European Community's Seventh Framework Programme (FP7/2007-2013)/ERC grant agreement 610055 as part of the 'ice2ice' project. The [10]Be measurements in the WAIS Divide core were funded by NSF grants ANT 0839042 (to M.W.C) and 0839137 (to K.C.W). Additional funding by the Columbia Climate Center of Columbia University (to F.M and W.J.D.) and the Vetlesen Foundation (W.J.D) is gratefully acknowledged. This study is a contribution to the INTIMATE project.

## Author contributions

F.M. conceived and designed the study, and wrote the first draft of the manuscript. W.J. D. contributed to the quantification and evaluation of biomarker data. A.S. designed and performed the climate model experiments. T.J.H. developed the Random Walk Model. N. L.B. performed the tephra analyses. N.D. together with F.M performed the biomarker analyses. M.W.C., T.E.W., and K.W measured the WAIS Divide Ice Core [10]Be data. L.C. S. provided complementary [14]C data for core MD99-2284. M.H.S. provided insight into regional paleoenvironment. T.M.D provided the background data for core MD99-2284 and together with F.M and W.J.D. secured necessary financial support. All authors contributed to the interpretation of the results and editing of the manuscript.

## Additional information

**Competing interests:** The authors declare no competing interests.

