## [Peer Review File · Nature Communications]

Reviewer #1 (Remarks to the Author):

A very interesting and potentially important study by Muschitiello et al. In my view, this would make a very important contribution if several issues can be adequately addressed.

Because so much of this contribution will hinge on the robustness of the age modeling and estimates of age uncertainty, I would strongly encourage that at least two experts be asked to assess the treatments presented in appendix 1.

I believe my own comments/suggestions can be addressed if the age modeling is sufficiently constrained. But I emphasize that when calling for a lead of AMOC changes to atmospheric temperature changes over Greenland on the scale of 4 centuries really requires that the AMOC records used in this study have unquestionable age models. I suspect that the reported age uncertainty estimates (e.g. page 5) of +/- 32 years is not a realistic estimate of the true timing uncertainty. This concern is particularly relevant to the argument for a AMOC transition leading into GI-1e.

The age uncertainty aside, there is another problem with the lead of AMOC strengthening going into GI-1 because there is only one benthic ^{14}C age between 16.0 and 14.5kyBP. Hence, if that single benthic ^{14}C age is an outlier and not truly representative of the age of deep waters at that time, then this assertion of changing B-P ages is also wrong. It is surprising to see only one benthic ^{14}C age in this critical interval. Is this because benthic forams are so rare in these intervals? And was this lowermost benthic age based upon *Pyrgo murrhina*? If that is the case, then it is even more questionable how representative and robust the b-p age actually is prior to GI-1.

Reading the methods section, I came away wondering the authors could be sure that they have identified all outliers. Apparently 7 outliers were identified and removed. But the methods do not state where in the core these so-called outliers occur. There appears to be a persistent occurrence of "secondary" effects. Despite the high sedimentation rate, there is clearly a problem getting enough 'well-preserved' benthic forams from this core. Consequently, inferring a 1000yr trend from a single benthic age prior to GI-1 is not compelling, in my opinion. And while calling upon the other records from the North Atlantic may be helpful, those records have perhaps even larger age uncertainties, which are not addressed in this study.

With only one 'apparent' good benthic ^{14}C age in the pre-GI-1 interval and looking at the uncertainties on the benthic R values shown in Figure 2 for example, may mean there was NO change in the benthic age between 16 and 14.6kyBP. On this basis, and given the questions about 'secondary' effects, it seems to me that at least one and preferably two more benthic ^{14}C ages in that critical 1000yr gap interval is necessary to validate the assertion that there was a AMOC lead to the GI-1 warming. The planktonic ^{14}C ages alone don't suffice.

This concern is carried forward to the authors' assertion that there was "rapid strengthening of deep vertical mixing of the well-ventilated, young surface waters with poorly-ventilated, old bottom waters that resided in the deep Nordic Seas prior to invigoration of NADW formation..." This seems to be a reasonable interpretation but here too the robustness of the surface reservoir age calculation is. This portion of the reservoir age estimation has much larger uncertainties due to both the lack of ash layers at those depths and the correlation uncertainties tied to the terrestrial records. The radiocarbon ages themselves have very large uncertainties in the 16-15kyBP interval and

importantly, the terrestrial records do not extend before 15.2kyBP. So again, there is no strong age control for the interval prior to 14.5-15.0kyBP, which is the critical time interval for assessing whether there was a MOC lead to the abrupt warming over Greenland at GI-1.

On page 4, another assertion that is not well-constrained by the benthic $\delta^{18}\text{O}$ data. The authors state: "This is supported by an abrupt +1.2‰ increase in benthic $\delta^{18}\text{O}$ values at our site, suggesting formation of deep cold waters, just prior to the start of GI-1". Here too there is a critical dependence on a single (?) benthic measurement for the intervals prior to 14.5kyBP (Supplemental Figure 7c). The supplemental figure plots the benthic $\delta^{18}\text{O}$ at their site as a line graph instead of showing data points so it is difficult to know how well constrained this benthic $\delta^{18}\text{O}$ change actually is. I certainly understand the motivations for this assertion and if valid would be quite important to the argument. But I personally believe that relying on a single benthic measurement to constrain an interpretation that involves 1000 years is dangerous, especially if benthic forams are critically rare in those intervals. Some reworking or residual benthics at the sample horizon would mean the values are not indicative of the time interval around 15.5kyBP. Why just measure 1 sample in those critical pre-GI-1 intervals? As is, there is a serious data gap prior to the onset of GI-1. And calling upon other data records is helpful as the authors do, but the age uncertainties in those records are not addressed.

It would be my recommendation that the authors fill in more benthic data in the pre-GI-1 to make their argument compelling. I hope the authors can adequately address the benthic age and oxygen isotope questions I raise here. If they can, their paper would be quite useful, and their final paragraph would be much more impactful.

Reviewer #2 (Remarks to the Author):

Review

My overall impression of this manuscript is that it merits eventual publication in a high profile journal like Nat Comm. If correct the results are novel and would be of broad interest to people working on past and present climate, and modeling. I think it has the potential to be very influential.

Based on a new synchronization that places initial warming at a core site in the Faeroe Shetland Channel before the abrupt Bolling warming in Greenland, and radiocarbon data from the same core, it appears that benthic-planktonic or benthic-atmospheric age variations may also change before the abrupt warming in Greenland. The authors attribute the radiocarbon variations to variations in

NADW formation (rate?). This implies that NADW formation leads abrupt climate change over Greenland. A similar lead is inferred into the YD cooling. If correctly synchronized and interpreted, these results have implications for the links between NADW, northward heat transport, and high latitude temperatures.

This paper presents planktonic and benthic ^{14}C dates (among lots of other data) from a sediment core at 1500 m in the Faeroe Shetland Channel (FSC) one of the two major passages for Nordic overflows to enter the north Atlantic. The premise is that convection ^{14}C to the deep waters that then make their way over the sills hence "NADW formation". The more convection, the younger the deep water relative to the surface and presumably the more NADW "forming" – that is, entering the North Atlantic. This is a sensible assumption, and this is a unique data set in terms of location, quality and resolution.

Generally speaking, I found parts of the discussion difficult to follow for three reasons. The first is the discussion in terms of GS and GL events. Most of the community is more familiar with the terms YD, BA, and HS1 are more easily envisioned. Understandably, the BA is variable, and each of the warmer and colder intervals have their own names which may go beyond the generalist's lexicon, and so a numbering scheme may seem reasonable. However I strongly encourage the use of the terms YD, BA, HS1 or H1, with the GS, GI etc in parenthesis. The second is that there is a tremendous amount of information in the SI, and the reading of much but not all of that material is both essential to understanding the conclusions/paper and distracting. The third is that the Discussion section did not tie directly back to the radiocarbon data in a way that I could understand.

GENERAL RECOMMENDATION: I recommend returning to authors for revision. They should focus on clarity - clarifying what controls the benthic ^{14}C at their site (and the P-B). In that regard, Sup figure 8 is instructive and they might make more use of it. A rapid onset of convection causes an abrupt aging of surface water then a gradual relaxation to younger but still old waters. The deep gets younger more gradually and asymptotes. Are these different responses consistent with the interpretations or are the interpretations too simplistic? Is it also possible to depict how the idealized Nordic Sea surface and deep respond to a shutdown?

Specific comments:

Paragraph beginning on line 74 to be clarified:

Unclear if implying that NADW formation moved south of Iceland-scotland ridge and if so how the Nordic seas could be a continuous source of NADW as stated, unless a very minor source. Or is it meant that it moved south within the Nordic Sea

82 not clear what "sign and signature" refers to.

86 here might make the link between convection, radiocarbon, and "NADW production" more explicit. Might also refer more directly to sup fig 8. Might also give a sentence about controls on surface radiocarbon. Later, important controls for both surface and deep are mentioned in passing.

Age model –The key to the conclusions is how well the records from different archives are correlated. The Intro line 60-64 suggests that previous results are somewhat ambiguous because of methods of synchronization, but the synchronization is as important for this study as for any other. A paragraph of methods, figure captions, and an extensive SI text suggest that the age models are reasonable. In sum while overwhelming, the age models and synchronization seem reasonable. Much of the results hinge on the assumption that temperature at the site remains correlated to the hydroclimate record even across abrupt transitions. This places the BA-associated warming at this site earlier than the abrupt warming in GL (Fig S6).

Can Figure 1 be extended to include GL air temperatures? Can the authors include the time series of core site SST and Swedish lake hydrology, in Loveclim and CCSM3 across the deglaciation to show there are no leads or lags across the major transitions e.g. consistent with the synchronization?

Line 104 – it is not clear to me how the records in Supp figures 6 and 7 reflect inflow and overflow of waters. If this figure is shown, it needs to be explained somewhere. Acknowledging that time scale can be an issue as discussed in the text or figure captions, it is also likely that there are real differences among sites. Since I don't fully understand the point of these figures I can't comment on whether that's important.

Line 106 “shortly preceding the warming transition” ...in Greenland - might add “abrupt “warming. Previous studies suggest some warming beginning before the abrupt HS-BA transition and as early as 20000 a. <https://doi.org/10.1016/j.crte.2005.05.011>

Also note that the 14C change is approximately synchronous with the warming transition in the same core.

Given the gap in the benthic record and the error on the second oldest (most relevant) point plotted, is it really clear that the B-P decrease occurred earlier than in GL?

Line 112 It is not clear to me why benthic d18O would increase when younger, presumably less dense, surface water is mixed into the deep water. It seems that if d18O increases as observed, then the 14C would be older (not observed). Please clarify the mechanism/explanation for observation.

Line 118 Can the authors please clarify why a southward migration of the sea ice margin and location of NADW production would result in younger deep waters. Is this because NADW is no longer tapping into deep Nordic Sea waters as it has moved south of sills? If so, if NADW is not largely formed in the Nordic Seas what is 14C at the FSC core site monitoring?

Understanding why the B-P age was smallest at ~13300 is key to interpreting the results and the benthic aging that follows. Can the trends be discussed in the context of Supp figure 8 and other potential factors? As is I don't feel I have a good understanding of what the authors believe controlled the deep water changes. Could the youngening trend reflect that the first strong convection flushes out more older water but continued convection and overflow would increasingly include less old water as there is increasingly less old water to mix in with young water as the Nordic Sea is "flushed", or some similar process? Such a process would decouple the 14C from NADW "formation". The reason I ask is that it is counterintuitive that there is more NADW formation, presumably more heat transported northward moving forward in time across the B-A, during which GL is cooling. Thus if this record represents NADW formation (and presumably export?) there is a strong decoupling between AMOC and heat transport, and some other process is causing GL to cool.

135 I agree with that the decline in P-14C between 13500 and 12900 could be due to a decrease in Atlantic Inflow to the Nordic Seas and greater equilibration with atmosphere, but then if the P-B record is a record of NADW production there is a decoupling between inflow and production. Can the deep aging occur in the absence of NADW weakening (see Supp figure 8)

Figure 3 shows that the B-atm trends first introduced in the figure are similar to trends documented elsewhere. Unlike Figure 2d, this figure only has data where benthic data exist, a plus. As such, it shows that the timing of the change at the HS1-BA boundary is within error of the sharp warming at GRIP (not shown). If the age models for the NE Atl cores are reliable then those data seem to support an earlier transition as proposed in this manuscript (the other sites are less clear). The NW and Equatorial data support the early transition from the BA to the YD. The reason for these similarities is not discussed. Do the authors think these parallel changes reflect ventilation at the sites or changes in the 14C of the north Atlantic source? If the FSC 14C records changes in convection in the Nordic Seas or N-S migration of the production site, why would we expect similar records at these other sites. I assume not an artifact of the Atmospheric 14C record which is common to all?

Lines 162-220 Most of the remainder of the manuscript draws on published data and concepts to explain the lag between high latitude NADW records and Greenland temperatures. While reasonable, I think it would be useful to tie the mechanisms more clearly to 14C rather than eg. Line 174 "support our observations". How is the timing compatible (line 175). Exactly how do these 14C records speak to the mechanisms? The bottom panels of Fig 3 are not called out or discussed. Do they support the mechanisms in the Discussion section?

Can points where benthic data exist be highlighted somehow (marked with superimposed open symbol perhaps) so that the reader can see where data really exist (Fig Fig 2d , 4c)?

There are lots of new data in the SI that are barely discussed. Is this the best place for them?

Site symbols in Figure S1 are very difficult to find.

SUMMARY;

The new 14C data are of high quality, the synchronization is almost compelling (I'd like to see lead-lag across transitions in addition to correlation maps, Fig S1). If the authors could improve clarity on details of interpretation as well as big picture dynamics this could be an excellent contribution for Nature Comm. If the interpretation holds, that NADW leads large and abrupt climate changes at high northern latitudes, this would be an important and paradigm-shifting result.

Reviewer #3 (Remarks to the Author):

Review by Anders Svensson of manuscript entitled 'Deep-water circulation changes lead North Atlantic climate during deglaciation' submitted to Nature Communications by Muschitiello et al. (manuscript NCOMMS-18-30280)

This manuscript (MS) covers a comprehensive amount of work. In fact, several of the figures in the extensive supplementary section cover topics that could make it out for individual studies. The MS appears already to be very carefully worked through, and I do not have any specific comments on technical aspects.

In this review, I will focus on the dating part of the manuscript (MS), which I believe makes up an essential part of the study, and which also happens to be my main area of expertise within this topic. Many of the conclusions of the MS build on the chronology of the MD99-2284 marine sediment core that is obtained in a somewhat controversial manner.

The dating of the marine sediment core is based on linking to the recently obtained 'downwind' Atteköp lake sediment record from southern Sweden. For the younger part of the cores, the linking is obtained via conventional tephra synchronization using four distinct tephra layers including the widespread and well-known Saksunarvatn and Vedde reference horizons. For the section of the records older than ca 12.6 ka, however, the marine and terrestrial cores are 'linked-by-proxies' using an advanced version of the technique some would refer to as wiggle-matching. The result of this exercise is shown in figure S2.b that compares the marine sediment sea surface temperature reconstruction with the leaf-wax dD record from the lake record that is interpreted as a marine moisture-source proxy. The linking of the two records allows for a transfer of the calibrated atmospheric C-14 lake record chronology to the marine record. Having obtained this calendar age scale for the marine record allows to extract the reservoir ages of planktic and benthic records

independently, which, in turn, allows for all of the subsequent discussion of leads, lags, and ocean ventilation of the Nordic seas.

So, the acceptance of the very exciting interpretation of records in the MS critically comes down to whether one 'believes' in the proxy-matching presented in Figure S2b or not. To my knowledge, it is common not to accept a such a linking approach, and I do know of several colleagues that would probably not accept it. For good reasons: Experience shows that the wiggle-matching approach has lead to many false conclusions because one has a tendency of seeing the patterns that fit the hypothesis even if the hypothesis is wrong. Furthermore, experience also shows that lake sediment records are notoriously difficult to date accurately and that they are often strongly influenced by the local environment. I trust the Attekölp record to be a well-dated high-quality record, but still it is a new record that needs time and more vigorous comparison to other records from the region before it can be established as a key record.

Nevertheless, I support the linking and thereby the conclusions made in this study. For the following reasons:

- The linking presented in figure S2b does look convincing to the naked eye. When I first browsed the MS, I wrongly assumed that the marine and the terrestrial records shown in the figure were from the same archive simply because they do look similar.

- The inferred marine - ice-core offset at the onset of GS-1 is of the same order of magnitude as the offset identified at the Holocene onset. The Holocene onset match is tephra-based and therefore a lot more solid. Of course there is no guarantee that there has to be offsets of similar magnitudes at both transitions, but it seems likely. It is disturbing that there are no tephras in the older part of the records and at the same time there is no similar good looking 'wiggle match' between the marine and the terrestrial records in the younger part.

The tie-point matching presented in Figure S4 I find less convincing than the linking presented in Figure S2. If I got it right, this matching is, however, only applied in the MS to estimate uncertainties of the match presented in Figure S2, and therefore it is not as critical for the conclusions of the study.

The ice core – C-14 synchronization presented in Figure S5 generally looks convincing. It is an approved technique for ice core (10Be) – dendrochronology (14C) synchronization and the derived offsets in chronologies of about +/-50 years agrees with those identified in other studies, such as the recent study by Adolphi et al., *Climate of the Past*, 2018. The Greenland 100 years 'jump' at 12.3 ka appears wired and unrealistic – if the course is a wrong tree-ring match in IntCal, it should also show up in the WD comparison? – but since all age scales show consistency within 100 years, there is no way those disagreements can generate 'fake' offset of time scales of about 400 years such as those identified in the present study.

In summary, I 'buy' the chronology part of the story, despite the somewhat controversial marine-terrestrial 'wobble-matching' approach. It would be convenient with another common tephra layer in the older part of the investigated time interval to verify the record alignment, but as that is currently not available, I think the MS offers the second best approach which is still convincing. I would not be surprised if later studies will be able to adjust the offsets identified in the present study, but in the deglacial period, offsets of the order of 400 yrs are too large to vanish with improved dating techniques, because the applied key chronologies are already in agreement within 100 years or better.

First, we would like to thank the reviewers for the time they took to provide a comprehensive review of our manuscript and for their constructive criticism. Their comments have helped us to craft a revised version of our study, as well as to re-examine some of our original interpretations. We particularly appreciated that each reviewer found our manuscript worth consideration for publication and recognized the importance of our data set. In response to the reviewer comments, we have made substantial changes to the main text and supplementary information. Please find below a detailed point-by-point response to the Reviewers' comments.

Reviewers' comments:

#####

Reviewer #1 (Remarks to the Author):

A very interesting and potentially important study by Muschitiello et al. In my view, this would make a very important contribution if several issues can be adequately addressed.

Because so much of this contribution will hinge on the robustness of the age modeling and estimates of age uncertainty, I would strongly encourage that at least two experts be asked to assess the treatments presented in appendix 1.

I believe my own comments/suggestions can be addressed if the age modeling is sufficiently constrained. But I emphasize that when calling for a lead of AMOC changes to atmospheric temperature changes over Greenland on the scale of 4 centuries really requires that the AMOC records used in this study have unquestionable age models. I suspect that the reported age uncertainty estimates (e.g. page 5) of +/- 32 years is not a realistic estimate of the true timing uncertainty. This concern is particularly relevant to the argument for a AMOC transition leading into GI-1e.

Thank you for bringing this up. This is a fair comment. The apparently small age uncertainties stem entirely from the Monte Carlo piecewise regression approach adopted here, which accounts for both analytical and chronological uncertainties associated with the proxy reconstructions. To our knowledge this is the most suitable and widely used technique to estimate duration and timing of transitions in climate-proxy series. Also, it should be noted that the uncertainties are expressed as 1 sigma errors. We apologise for not having specified this earlier. We now clearly state that the uncertainties are a result of the statistical approach used here and that the uncertainties encompass 1 sigma (new lines 102 and new lines 174).

The age uncertainty aside, there is another problem with the lead of AMOC strengthening going into GI-1 because there is only one benthic 14C age between 16.0 and 14.5kyBP. Hence, if that single benthic 14C age is an outlier and not truly representative of the age of deep waters at that time, then this assertion of changing B-P ages is also wrong. It is surprising to see only one benthic 14C age in this critical interval. Is this because benthic forams are so rare in these intervals? And was this lowermost benthic age based upon *Pyrgo murrhina*? If that is the case, then it is even more questionable how representative and robust the b-p age actually is prior to GI-1.

Thank you for your comment. We agree that our record lacks sufficient data to allow a solid interpretation of the benthic ventilation age prior to GI-1. We would like to reassure Reviewer1 that the one benthic age available is not based on *Pyrgo* spp –which we do not use in our study.

We have repeatedly, but unsuccessfully attempted at sampling this important interval. Regretfully our search over the interval spanning ~14.6-15.0 kyr BP only resulted in four datable samples: one sample is presented in Figure 1; one is a specimen of *Pyrgo murrhina* (not used for the construction of our ventilation record); two very small samples (i.e. <0.5mg) yielded anomalously too young ages probably due to modern carbon contamination. This is also confirmed by acid-leaching tests performed on these samples, which indicate likely contamination by adsorption of modern atmospheric CO₂. We apologise that this information was not provided in the manuscript. This is now specified in new lines 365-377. A spreadsheet with detailed information on 14C dating results and outlier selection will be provided as a Supplementary Dataset upon publication.

Reading the methods section, I came away wondering the authors could be sure that they have identified all outliers. Apparently 7 outliers were identified and removed. But the methods do not state where in the core these so-called outliers occur. There appears to be a persistent occurrence of “secondary” effects. Despite the high sedimentation rate, there is clearly a problem getting enough ‘well-preserved’ benthic forams from this core. Consequently, inferring a 1000yr trend from a single benthic age prior to GI-1 is not compelling, in my opinion.

We agree with the Reviewer and have now provided information on the position of the outliers downcore (new lines 362-363). The secondary effect seems to be mainly associated with contamination by modern carbon via adsorption of CO₂, which is particularly critical when dealing with very small/fine samples (e.g. Nadeau et al., 2001; Sepulcre et al., 2017).

And while calling upon the other records from the North Atlantic may be helpful, those records have perhaps even larger age uncertainties, which are not addressed in this study. With only one ‘apparent’ good benthic 14C age in the pre-GI-1 interval and looking at the uncertainties on the benthic R values shown in Figure 2 for example, may mean there was NO change in the benthic age between 16 and 14.6kyBP. On this basis, and given the questions about ‘secondary’ effects, it seems to me that at least one and preferably two more benthic 14C ages in that critical 1000yr gap interval is necessary to validate the assertion that there was a AMOC lead to the GI-1 warming. The planktonic 14C ages alone don’t suffice. This concern is carried forward to the authors’ assertion that there was “rapid strengthening of deep vertical mixing of the well-ventilated, young surface waters with poorly-ventilated, old bottom waters that resided in the deep Nordic Seas prior to invigoration of NADW formation...” This seems to be a reasonable interpretation but here too the robustness of the surface reservoir age calculation is. This portion of the reservoir age estimation has much larger uncertainties due to both the lack of ash layers at those depths and the correlation uncertainties tied to the terrestrial records. The radiocarbon ages themselves have very large uncertainties in the 16-15kyBP interval and importantly, the terrestrial records do not extend before 15.2kyBP. So again, there is no strong age control for the interval prior to 14.5-15.0kyBP, which is the critical time interval for assessing whether there was a MOC lead to the abrupt warming over Greenland at GI-1.

We agree that the other marine sediment records presented here (specifically data from Ezat et al., 2017 and Thornalley et al., 2015) are probably affected by very large age uncertainties. Notably, we did not find any clear assessment of the chronological errors in these two studies (i.e. the 14C dates are virtually perfectly placed in time). We have therefore conservatively applied +/- 200 years (1 sigma) uncertainty to each 14C measurement (please note error bars in Figure 3). These error bounds do not necessarily reflect the true uncertainty of the underlying chronologies but certainly seem like a conservative estimate.

Quantifying the actual extent of such uncertainties was beyond the scope of our study. Rather, our aim was to compare our data to other well-dated marine records, such as the deep-sea coral records presented in Figure 3. Nonetheless, to support our B-P estimate prior to GI-1, we now provide a new correlation of two Norwegian Sea coring sites (JM-FI-19PC in the Faroe-Shetland Channel –1200m– and MD95-2011 on the Vøring Plateau –1000m) to MD99-2284 using *N. pachyderma* (*s*) $\delta^{18}\text{O}$ (i.e. the record with the best sampling resolution in MD99-2284). The results are presented in new Supplementary Figure 9. The $\delta^{18}\text{O}$ records are easily correlatable around the HS1/BA transition and downcore, and feature shifts of similar magnitude. The two records are accompanied by a number of benthic ^{14}C dates that clearly indicate older bottom ^{14}C ages before the onset of GI-1 (Bølling warming) that are in broad agreement with our one pre-GI-1 benthic date (Supplementary Figure 9b). In particular, these independent records show B-P ages of 1000-1500 years, which is in line with our results presented in Figure 2d. Especially, three bottom ^{14}C dates were obtained from a solitary deep-sea coral recovered in core MD95-2011, which was absolutely dated using U/Th (Dreger, 1999) at ~16.0 kyr BP. The three bottom ^{14}C ages are stratigraphically accompanied by one surface ^{14}C age based on *N. pachyderma* (*s*), resulting in a B-P offset of ~1200 years. This finding is robust and is now presented in the main Figure 2d as supporting evidence for our pre-GI-1 large B-Atm offset.

In addition, we toned down the claim that the decrease in B-P ages occurs before the onset of GI-1 (new line 108). We also specify that our interpretation is only based on one B-P estimate and should be therefore picked with a grain of salt (new lines 112-114).

Even though we now provide new supporting evidence, and agree with Reviewer1 that it is important to fill the data gaps before the onset of GI-1, we would like to stress that the pre-GI-1 interval is not the focus of our study, which primarily deals with the mechanisms surrounding the transition into and out of the YD.

On page 4, another assertion that is not well-constrained by the benthic $\delta^{18}\text{O}$ data. The authors state: “This is supported by an abrupt +1.2‰ increase in benthic $\delta^{18}\text{O}$ values at our site, suggesting formation of deep cold waters, just prior to the start of GI-1“. Here too there is a critical dependence on a single (?) benthic measurement for the intervals prior to 14.5kyBP (Supplemental Figure 7c). The supplemental figure plots the benthic $\delta^{18}\text{O}$ at their site as a line graph instead of showing data points so it is difficult to know how well constrained this benthic $\delta^{18}\text{O}$ change actually is. I certainly understand the motivations for this assertion and if valid would be quite important to the argument. But I personally believe that relying on a single benthic measurement to constrain an interpretation that involves 1000 years is dangerous, especially if benthic forams are critically rare in those intervals. Some reworking or residual benthics at the sample horizon would mean the values are not indicative of the time interval around 15.5kyBP.

We fully agree that our claim is stretched too far. We have now toned down our claims of a lead prior to the onset of GI-1 (new line 108 and new lines 112-114). Also please note that following comments from Reviewer2, we have now substantially changed the Results section, as well as the interpretation of the sequence of events, and we no longer discuss the shifts in benthic $\delta^{18}\text{O}$ around the onset of GI-1 (new lines 103-126).

Why just measure 1 sample in those critical pre-GI-1 intervals? As is, there is a serious data gap prior to the onset of GI-1. And calling upon other data records is helpful as the authors do, but the age uncertainties in those records are not addressed.

Unfortunately, the lack of well-preserved benthic foraminifera in the interval preceding GI-1 has made it difficult to generate a continuous record of benthic $\delta^{18}\text{O}$ at our site (we only have two data points prior to the start of GI-1).

Anyway, regardless of whether we discuss the timing of benthic $\delta^{18}\text{O}/^{14}\text{C}$ or not, we would like to point out again that the pre-GI-1 patterns are not the focus of our study and hence do not undermine our main conclusions, i.e. that Nordic Seas intermediate/deep water changed out of phase with Greenland climate at the transitions into and out of the YD. Further work will be needed to address the timing of NADW formation changes in the Nordic Seas relative to Greenland climate during HS-1 and LGM.

It would be my recommendation that the authors fill in more benthic data in the pre-GI-1 to make their argument compelling. I hope the authors can adequately address the benthic age and oxygen isotope questions I raise here. If they can, their paper would be quite useful, and their final paragraph would be much more impactful.

#####

Reviewer #2 (Remarks to the Author):

Review

My overall impression of this manuscript is that it merits eventual publication in a high profile journal like Nat Comm. If correct the results are novel and would be of broad interest to people working on past and present climate, and modeling. I think it has the potential to be very influential.

Based on a new synchronization that places initial warming at a core site in the Faeroe Shetland Channel before the abrupt Bolling warming in Greenland, and radiocarbon data from the same core, it appears that benthic-planktonic or benthic-atmospheric age variations may also change before the abrupt warming in Greenland. The authors attribute the radiocarbon variations to variations in NADW formation (rate?). This implies that NADW formation leads abrupt climate change over Greenland. A similar lead is inferred into the YD cooling. If correctly synchronized and interpreted, these results have implications for the links between NADW, northward heat transport, and high latitude temperatures.

This paper presents planktonic and benthic ^{14}C dates (among lots of other data) from a sediment core at 1500 m in the Faeroe Shetland Channel (FSC) one of the two major passages for Nordic overflows to enter the north Atlantic. The premise is that convection ^{14}C to the deep waters that then make their way over the sills hence “NADW formation”. The more convection, the younger the deep water relative to the surface and presumably the more NADW “forming” – that is, entering the North Atlantic. This is a sensible assumption, and this is a unique data set in terms of location, quality and resolution.

Generally speaking, I found parts of the discussion difficult to follow for three reasons. The first is the discussion in terms of GS and GL events. Most of the community is more familiar with the terms YD, BA, and HS1 are more easily envisioned. Understandably, the BA is variable, and each of the warmer and colder intervals have their own names which may go beyond the generalist’s lexicon, and so a numbering scheme may seem reasonable. However I strongly encourage the use of the terms YD, BA, HS1 or H1, with the GS, GI etc in parenthesis.

Thank you for your comment. We agree that the community might not be fully familiar with the Greenland Event Stratigraphy. However, the main goal of our study is to compare timing of ocean circulation changes to climate events recorded in Greenland ice cores. Therefore we deem appropriate to use the specific terminology for the isotopic shifts observed in Greenland rather than the generic deglacial terminology. The widely used terms YD and BA were originally coined to define pollen-stratigraphic boundaries in southern Scandinavia (i.e. von Post, 1916) and not climate transitions in Greenland. Similarly, H1 is a strictly paleoceanographic term, defining an interval the boundaries of which change from site to site. For instance, recent work has clearly demonstrated that there is a significant lag between the onset of GS-1 in Greenland and the YD biostratigraphic boundary in Europe (e.g. Muschitiello et al., 2015; Rach et al., 2014). In order to avoid confusion and overlapping terminology we would prefer to stand by the correct wording of the events we are dealing with in our study, i.e. Greenland climatic events.

Nonetheless, to aid the reader we have now changed all the figures and associated captions to include the terms YD, BA and HS1 *sensu lato* (in parenthesis) and specifying that we refer to these events as stadials/interstadials. Where appropriate, we now also indicate these terms in parenthesis along the main text.

The second is that there is a tremendous amount of information in the SI, and the reading of much but not all of that material is both essential to understanding the conclusions/paper and distracting.

Thank you for bringing this up. This is a fair comment and we agree with the Reviewer. Accordingly, we have now moved a substantial portion of the Supplementary Material to the Methods section along the main text so that all the essential methodological information are now more readily available to the reader. Specifically, we now provide in the new Methods additional information regarding chronology, tephrochronology, ^{14}C dating of foraminifera, ΔR modelling, and timescale synchronization. The new Methods is shorter than 3000 words and therefore meets the journal guidelines for this section.

The third is that the Discussion section did not tie directly back to the radiocarbon data in a way that I could understand.

Please see replies to the specific comments on the Discussion below. We have now addressed the Reviewer's points regarding clarity on the controlling factors of our benthic ^{14}C .

GENERAL RECOMMENDATION: I recommend returning to authors for revision. They should focus on clarity - clarifying what controls the benthic ^{14}C at their site (and the P-B). In that regard, Sup figure 8 is instructive and they might make more use of it. A rapid onset of convection causes an abrupt aging of surface water then a gradual relaxation to younger but still old waters. The deep gets younger more gradually and asymptotes. Are these different responses consistent with the interpretations or are the interpretations too simplistic?

We apologise for the confusion and thank Reviewer2 for giving us the opportunity to revise and clarify our interpretations. We now more straightforwardly argue that bottom ^{14}C ventilation at our site reflect primarily deep convection and southward deep-water export (new lines 86-87). We deem this as the most conservative and widely adopted interpretation of benthic ^{14}C records in this region (e.g. Ezat et al., 2017; Thornalley et al., 2015). This interpretation is confirmed by the strong similarity between our B-Atm records and the atmospheric (production-corrected) $\Delta^{14}\text{C}$ data, as well as other B-Atm records downstream

(Fig. 3), thus suggesting that our B-Atm (and by extension B-P) reconstructions are likely monitoring a basin-wide signal associated with NADW production rate and export. We now explicitly state this in the new Discussion (new lines 165-172). Furthermore, we provide a new supplementary figure (Supplementary Fig. 11) where we present results from transient climate simulations of an AMOC shutdown and recovery, which demonstrate a tight coupling between changes in AMOC and B-P age offsets in the Norwegian Sea. Furthermore, the model results support our interpretation that deep ventilation in the Norwegian Sea is strongly related to large-scale NADW circulation.

As to the fact that deeper water becomes younger more gradually than the surface at the onset of GI-1, we have now carefully re-evaluated this interpretation and concluded that it is not likely for two reasons.

1) P-Atm ages increase by 500 years at the onset of GI-1, but rather than being accompanied by a concomitant decrease in B-Atm ages, bottom ventilation equally increase by 500 years (Fig. 3c). This is at odds with our initial interpretation and results from our 1-D column model experiment. It is also in contrast with our new model results presented in Supplementary Figure 11, which show that NADW/AMOC recovery results in a rapid aging of P-Atm and youngening of B-Atm (i.e. a decoupling between surface and bottom ^{14}C ages). Hence, the covariability pattern observed in our B-Atm and P-Atm reconstructions at the onset of GI-1 disproves a rapid strengthening of deep vertical mixing.

2) Climate model experiments (Cheng et al., 2011; Renold et al., 2009) point at a ~400-year long resumption (~400 years) of AMOC at the transition out of cold stadial conditions, in particular at the onset of GI-1 (Bølling warming) (Cheng et al., 2011). This slow AMOC resumption, which is robust across models, involves a gradual northward recovery of NADW formation in the North Atlantic, characterised by a late re-initiation of deep convection in the Nordic Seas relative to other more southerly located deep convection sites (e.g. Labrador Sea).

Based on these lines of evidence, we have rectified our interpretation of the factors controlling our benthic ^{14}C record across the onset of GI-1 (Bølling warming) and now argue that the gradual trend towards lower B-P offsets is an expression of a time-transgressive northward re-initiation of NADW production in the Nordic Seas with respect to deep convection sites south of Greenland. Moreover, our new interpretation of a delayed re-activation of deep convection in the Nordic Seas after the start of GI-1 is in good agreement with paleoceanographic reconstructions from the deep subarctic Nordic Seas, showing that modern-like deep-water conditions first developed midway through GI-1 (Bølling-Allerød) or a few centuries before the YD (Bauch et al., 2001). This is all explained in detail in the new Results section (new lines 116-126). As a result of the new interpretation we have removed any reference to the 1-D column model experiment and deleted old Supplementary Fig. 8.

Please also note that we have re-assessed the timing of changes in B-P offset in our data taking into consideration the broad ^{14}C uncertainties. Specifically, we now argue that B-Atm ages and B-P offsets reach near-modern values earlier than previously proposed (i.e. as early as 14.2 kyr BP – see Fig. 2c-d) (new line 116). This timing implies a ~400 year lag between the start of GI-1 (Bølling warming) and the establishment of modern-like deep convection conditions in the Nordic Seas, which is also more in line with the duration of AMOC resumption observed in the climate model studies discussed above.

Is it also possible to depict how the idealized Nordic Sea surface and deep respond to a shutdown?

We now provide a new figure (Supplementary Fig. 11), as detailed in my previous comment, showing the deep Norwegian Sea ventilation response to AMOC shutdown and subsequent resumption using a transient climate model simulation with the UVic model (Supplementary Information). The results reveal a strong consistency between modelled and reconstructed B-P offsets (both in sign and magnitude), supporting our view that our benthic ^{14}C records reflect North Atlantic wide weakening of NADW formation. In particular, the simulated P-Atm and B-Atm offset are in excellent agreement with our reconstructed planktic and benthic ^{14}C data. Notably, a clear decoupling between surface and bottom ^{14}C age is observed at the transitions into and out of the simulated stadial, which is consistent with the patterns identified in our records (see Fig. 2c). The decoupling is indicative of a more prominent water-column stratification, whereby shoaling of the surface mixed layer results in enhanced pCO_2 equilibration between surface waters and the atmosphere, whereas isolation of the deep layer hinders ventilation and leads to older bottom ^{14}C ages via radioactive decay.

For reference, below we present results from our 1-D column diffusion model (as initially requested by Reviewer2) where we simulated a rapid reduction of deep-water formation by first running the model to steady-state conditions with high vertical diffusivity regime ($10^{-3} \text{ m}^2 \text{ s}^{-1}$), and subsequently running in forward mode with diffusivity regime that decreases exponentially from $10^{-4} \text{ m}^2 \text{ s}^{-1}$ to $10^{-5} \text{ m}^2 \text{ s}^{-1}$, equally to the setup used in old Supplementary Figure 8a. Panel **a**) shows the individual P-Atm and B-Atm disequilibrium ages, and **b**) shows the B-P offset (1500 m *minus* 100 m). It is worth to note that the P-Atm offset decreases towards younger ages (a few decades; hard to see due to scaling) at the start of the perturbation, similarly to results from the UVic simulation and to our proxy data at the onset of GS-1.

Specific comments:

Paragraph beginning on line 74 to be clarified:

Unclear if implying that NADW formation moved south of Iceland-scotland ridge and if so how the Nordic seas could be a continuous source of NADW as stated, unless a very minor source. Or is it meant that it moved south within the Nordic Sea

Thank you for bringing this up. This was confusing so we have now deleted the statement regarding the southward shift of the deep convection sites (new lines 77-87)—also considering the new interpretations discussed in our previous reply. In light of the good agreement between our deep convection data and other downstream records, we argue that the Nordic Seas indeed

remained a continuous source of NADW throughout the deglaciation, also in agreement with evidence from other regional reconstructions (Crocket et al., 2011; Meland et al., 2008).

82 not clear what “sign and signature” refers to.

This has been deleted.

86 here might make the link between convection, radiocarbon, and “NADW production” more explicit. Might also refer more directly to sup fig 8. Might also give a sentence about controls on surface radiocarbon. Later, important controls for both surface and deep are mentioned in passing.

We deem this paragraph sufficiently clear as to the link between deep 14C and NADW. Also, we would be reluctant to include a comment about the controls on surface radiocarbon as this is not relevant for this work (i.e. surface 14C is only mentioned –and explained– once and very briefly in the Discussion).

Age model –The key to the conclusions is how well the records from different archives are correlated. The Intro line 60-64 suggests that previous results are somewhat ambiguous because of methods of synchronization, but the synchronization is as important for this study as for any other.

Thank you for your comment. Our point is that marine records from the eastern sector of the Nordic Seas can be stratigraphically aligned to terrestrial records from the adjacent coastal areas more reliably (or at least with a fewer number of assumptions) than to Greenland ice cores, by virtue of the well-established link between coastal hydro-climate and upwind, near-field oceanographic metrics (e.g. Muschitiello et al., 2015). This is explained in detail in new lines 273-277. However, we concede that this is a climate synchronization and not a truly independent dating approach.

A paragraph of methods, figure captions, and an extensive SI text suggest that the age models are reasonable. In sum while overwhelming, the age models and synchronization seem reasonable. Much of the results hinge on the assumption that temperature at the site remains correlated to the hydroclimate record even across abrupt transitions. This places the BA-associated warming at this site earlier than the abrupt warming in GL (Fig S6).

We agree with the Reviewer that seemingly the SST warming occurs before the onset of GI-1. This however is not a genuine feature, but rather an expression of the age uncertainty of our lake sediment records (transferred onto MD99-2284). The figure does not convey the chronological uncertainty of the paleoceanographic records (i.e. only the median age values), which is up to a few centuries in this interval (see Supplementary Fig. 3c). This becomes clear when age uncertainty is shown explicitly such as in Fig. 3c (see error bars). We have now included a note in the caption of Supplementary Figure 7 to clarify that the rapid GI-1 warming in Greenland falls within the age uncertainty of the increase in SSTs recorded in core MD99-2284.

Can Figure 1 be extended to include GL air temperatures? Can the authors include the time series of core site SST and Swedish lake hydrology, in Loveclim and CCSM3 across the deglaciation to show there are no leads or lags across the major transitions e.g. consistent with the synchronization?

This is a very good point. Thank you for your comment. We now provide this important information in a new figure (Supplementary Fig. 2). As the Reviewer requested, we now present simulated temperature time series for NGRIP and for our coring site MD99-2284, and hydroclimate (specific humidity) for our terrestrial site Attekop. We also estimated the moving correlation between temperature and specific humidity at MD99-2284 and Attekop, respectively. The analysis shows that there is a significant positive correlation between hydrography at MD99-2284 and hydroclimate at Attekop throughout the deglaciation. Remarkably, both model results with LOVECLIM and CCSM3 indicate that the correlation is overall relatively stronger across the major transitions that we discuss in our study (i.e. GS-2/GI-1, GI-1/GS-1, and GS-1/Holocene). These new results further support our synchronization between SST and hydroclimate as reconstructed in MD99-2284 and Attekop, respectively.

Line 104 – it is not clear to me how the records in Supp figures 6 and 7 reflect inflow and overflow of waters. If this figure is shown, it needs to be explained somewhere.

We apologise for the lack of clarity. The compiled paleoceanographic records are located in the eastern Nordic Seas. The planktic records presented in Supplementary Figure 7 –but mainly in Supplementary Figure 8b– monitor the inflow of warm Atlantic surface/subsurface water into the Nordic Seas, whereas the benthic records presented in Supplementary Figure 8c track returning overflow waters across the Iceland-Scotland Ridge. The text has now been rephrased to improve clarity (new lines 103-106).

Acknowledging that time scale can be an issue as discussed in the text or figure captions, it is also likely that there are real differences among sites. Since I don't fully understand the point of these figures I can't comment on whether that's Important.

We now mention in the caption of Supplementary Figure 8 that the differences can be authentic time lags among sites.

Line 106 “shortly preceding the warming transition”...in Greenland - might add “abrupt “warming. Previous studies suggest some warming beginning before the abrupt HS-BA transition and as early as 20000 a. <https://doi.org/10.1016/j.crte.2005.05.011> Also note that the 14C change is approximately synchronous with the warming transition in the same core.

This has been edited accordingly (new line 108).

Given the gap in the benthic record and the error on the second oldest (most relevant) point plotted, is it really clear that the B-P decrease occurred earlier than in GL?

This is a good point also raised by Reviewer1 (please see my previous comments). We think we have now addressed this problem by toning down our claims and by explicitly state that our interpretation is based on only one data point (new line 108). We also provide new independent B-P age estimates from the Norwegian Sea in support of our pre-GI-1 (HS-1) 14C measurements (new Fig. 2d and new Supplementary Fig. 9, and new lines 112-114).

Line 112 It is not clear to me why benthic d18O would increase when younger, presumably less dense, surface water is mixed into the deep water. It seems that if d18O increases as observed, then the 14C would be older (not observed). Please clarify the mechanism/explanation for observation.

Benthic $\delta^{18}\text{O}$ values in this region can be primarily related to bottom water temperature (Rasmussen and Thomsen, 2004). Deep convective renewal of deep and bottom waters via loss of buoyancy –and hence mixing with relatively young surface water– is generally associated with thermal cooling, which results in an isotopic fractionation in benthic foraminifera shells towards the heavier ^{18}O isotope.

Please note that this portion of the Results section has been entirely removed and we no longer discuss benthic $\delta^{18}\text{O}$ records (new lines 103-126).

Line 118 Can the authors please clarify why a southward migration of the sea ice margin and location of NADW production would result in younger deep waters. Is this because NADW is no longer tapping into deep Nordic Sea waters as it has moved south of sills? If so, if NADW is not largely formed in the Nordic Seas what is ^{14}C is at the FSC core site monitoring?

Please see my previous comments. This issue has now been addressed and we no longer discuss NADW as a function of sea ice front migrations. We have now hopefully clarified the factors controlling changes in benthic ^{14}C at our site and interpret our records primarily as an expression of deep convection and NADW formation rates in the Nordic Seas (new lines 83-87).

Understanding why the B-P age was smallest at ~13300 is key to interpreting the results and the benthic aging that follows. Can the trends be discussed in the context of Supp figure 8 and other potential factors? As is I don't feel I have a good understanding of what the authors believe controlled the deep water changes. Could the youngening trend reflect that the first strong convection flushes out more older water but continued convection and overflow would increasingly include less old water as there is increasingly less old water to mix in with young water as the Nordic Sea is "flushed", or some similar process? Such a process would decouple the ^{14}C from NADW "formation". The reason I ask is that it is counterintuitive that there is more NADW formation, presumably more heat transported northward moving forward in tie across the B-A, during which GL is cooling. Thus if this record represents NADW formation (and presumably export?) there is a strong decoupling between AMOC and heat transport, and some other process is causing GL to cool.

Please see my previous comments. We now argue that the trend towards younger benthic ages throughout GI-1 is caused by a northward re-initiation of deep convection and NADW formation in the Nordic Seas rather than the limiting timing associated with evacuation of old carbon from the deep Nordic Seas. This is in line with mechanisms observed in transient climate model experiments simulating AMOC recovery at the transition into the Bølling interstadial (Cheng et al., 2011). It is also in agreement with independent carbon isotope records from the subarctic Nordic Seas, which suggest that deep convection started to reinvigorate relatively later than the onset of GI-1 (Bølling interstadial), and precisely a few centuries before the YD stadial (~13.5 kyr BP) (Bauch et al., 2001). This is now explained in detail in new lines 116-126.

As to the Reviewer's final comment, we fully agree that there is a decoupling between NADW and Greenland temperature. This is the key message of our study and highlights the need of ad-hoc modelling experiments to reproduce and understand the mechanisms behind this decoupling/lagged response.

135 I agree with that the decline in P- ^{14}C between 13500 and 12900 could be due to a decrease in Atlantic Inflow to the Nordic Seas and greater equilibration with atmosphere, but then if the

P-B record is a record of NADW production there is a decoupling between inflow and production. Can the deep aging occur in the absence of NADW weakening (see Supp figure 8)

Thank you for your comment. We believe this confusion arises from a wrong assignment of the events we discuss in the manuscript. We apologise for this. We have re-evaluated the timing of changes in B-Atm and P-Atm offsets during GI-1 taking into full account the analytical uncertainties (new line 116 and new line 144). A more careful scrutiny of our data set shows that a significant decrease in P-Atm ages (relative to the GI-1 mean) only occur after 13250 yrs BP (see Fig. 2c), and therefore there is no decoupling between the P-Atm and B-P offsets. Please note that the overall patterns of B-Atm and P-Atm values compare very well with results from our model simulations presented in the new Supplementary Figure 12, and suggests a clear imprint of NADW reduction/recovery in our 14C records.

Figure 3 shows that the B-atm trends first introduced in the figure are similar to trend documented elsewhere. Unlike Figure 2d, this figure only has data where benthic data exist, a plus. As such, it shows that the timing of the change at the HS1-BA boundary is within error of the sharp warming at GRIP (not shown). If the age models for the NE Atl cores are reliable then those data seem to support an earlier transition as proposed in this manuscript (the other sites are less clear). The NW and Equatorial data support the early transition from the BA to the YD. The reason for these similarities is not discussed. Do the authors think these parallel changes reflect ventilation at the sites or changes in the 14C of the north Atlantic source? If the FSC 14C records changes in convection in the Nordic Seas or N-S migration of the production site, why would we expect similar records at these other sites. I assume not an artifact of the Atmospheric 14C record which is common to all?

Thank you for your comment. As mentioned earlier we have toned down our claims regarding a potential lead of NADW ahead of GI-1. As to the 14C data comparison, we think that the similarity among the ventilation records reflect downstream transfer of the deep/intermediate 14C signal from the high latitude North Atlantic to lower latitudes. Also note that the similarity lends supports to our interpretation that B-Atm ages in the Nordic Seas are related to NADW circulation and likely the larger AMOC system. We now discuss these points in new lines 157-172.

Lines 162-220 Most of the remainder of the manuscript draws on published data and concepts to explain the lag between high latitude NADW records and Greenland temperatures. While reasonable, I think it would be useful to tie the mechanisms more clearly to 14C rather than eg. Line 174 “support our observations”. How is the timing compatible (line 175). Exactly how do these 14C records speak to the mechanisms? The bottom panels of Fig 3 are not called out or discussed. Do they support the mechanisms in the Discussion section?

We have now rephrased these lines accordingly (new lines 198-200 and new line 2013).

The histograms in Fig. 3 only show the depth of individual coral 14C measurements to demonstrate that trends are not an artefact of grouping coral data. This is indicated in the caption of Fig. 3.

Can points where benthic data exist be highlighted somehow (marked with superimposed open symbol perhaps) so that the reader can see where data really exist (Fig Fig 2d , 4c)?

We have now modified Fig. 4 so that it shows the position of individual benthic 14C measurements.

There are lots of new data in the SI that are barely discussed. Is this the best place for them?

Please see our previous replies. We have now moved a substantial portion of the Supplementary Methods to the Methods section along the main text.

Site symbols in Figure S1 are very difficult to find.

Site symbols have been added.

SUMMARY;

The new 14C data are of high quality, the synchronization is almost compelling (I'd like to see lead-lag across transitions in addition to correlation maps, Fig S1). If the authors could improve clarity on details of interpretation as well as big picture dynamics this could be an excellent contribution for Nature Comm. If the interpretation holds, that NADW leads large and abrupt climate changes at high northern latitudes, this would be an important and paradigm-shifting result.

#####

Reviewer #3 (Remarks to the Author):

Review by Anders Svensson of manuscript entitled 'Deep-water circulation changes lead North Atlantic climate during deglaciation' submitted to Nature Communications by Muschitiello et al. (manuscript NCOMMS-18-30280)

This manuscript (MS) covers a comprehensive amount of work. In fact, several of the figures in the extensive supplementary section cover topics that could make it out for individual studies. The MS appears already to be very carefully worked through, and I do not have any specific comments on technical aspects.

Thank you for your comment. Following the comments of Reviewer1 and 2 we have moved a large portion of the Supplementary Methods to the Methods section of the paper along the main text.

In this review, I will focus on the dating part of the manuscript (MS), which I believe makes up an essential part of the study, and which also happens to be my main area of expertise within this topic. Many of the conclusions of the MS build on the chronology of the MD99-2284 marine sediment core that is obtained in a somewhat controversial manner.

The dating of the marine sediment core is based on linking to the recently obtained 'downwind' Atteköp lake sediment record from southern Sweden. For the younger part of the cores, the linking is obtained via conventional tephra synchronization using four distinct tephra layers including the widespread and well-known Saksunarvatn and Vedde reference horizons. For the section of the records older than ca 12.6 ka, however, the marine and terrestrial cores are 'linked-by-proxies' using an advanced version of the technique some would refer to as wiggle-matching. The result of this exercise is shown in figure S2.b that compares the marine sediment sea surface temperature reconstruction with the leaf-wax dD record from the lake record that is interpreted as a marine moisture-source proxy. The linking of the two records allows for a transfer of the calibrated atmospheric C-14 lake record chronology to the marine record. Having obtained this calendar age scale for the marine record allows to extract the reservoir

ages of planktic and benthic records independently, which, in turn, allows for all of the subsequent discussion of leads, lags, and ocean ventilation of the Nordic seas.

So, the acceptance of the very exciting interpretation of records in the MS critically comes down to whether one 'believes' in the proxy-matching presented in Figure S2b or not. To my knowledge, it is common not to accept a such a linking approach, and I do know of several colleagues that would probably not accept it. For good reasons: Experience shows that the wiggle-matching approach has lead to many false conclusions because one has a tendency of seeing the patterns that fit the hypothesis even if the hypothesis is wrong. Furthermore, experience also shows that lake sediment records are notoriously difficult to date accurately and that they are often strongly influenced by the local environment. I trust the Atteköp record to be a well-dated high-quality record, but still it is a new record that needs time and more vigorous comparison to other records from the region before it can be established as a key record. Nevertheless, I support the linking and thereby the conclusions made in this study. For the following reasons:

- The linking presented in figure S2b does look convincing to the naked eye. When I first browsed the MS, I wrongly assumed that the marine and the terrestrial records shown in the figure were from the same archive simply because they do look similar.
- The inferred marine - ice-core offset at the onset of GS-1 is of the same order of magnitude as the offset identified at the Holocene onset. The Holocene onset match is tephra-based and therefore a lot more solid. Of course there is no guarantee that there has to be offsets of similar magnitudes at both transitions, but it seems likely. It is disturbing that there are no tephras in the older part of the records and at the same time there is no similar good looking 'wiggle match' between the marine and the terrestrial records in the younger part.

We thank Reviewer3 for his comments and we are pleased that he accepts our chronology. We would like to point out that, although Attekop is a relatively new site we published a comprehensive study and compared its records to other regional reconstructions (Wohlfarth et al., 2018). Together with other previously published work (Muschitiello et al., 2015) it has been demonstrated that paleo-hydroclimate data from this region are suitable for establishing marine-terrestrial synchronizations, as also revealed by our climate model analysis (Supplementary Fig. 1-2). We agree that an independent dating method is preferable to constrain the age of our marine records that avoids the use of climate wiggle matching. However, as also pointed out by the Reviewer, it is difficult to explain lags of the order of ~400 years between MD99-2284 and Attekop/Greenland, and it does not seem likely that further improvements to our chronology will explain such a large offset.

As to the last comment we would like to add that we believe that actually there is a potentially good match between the terrestrial and marine data in the upper part of the records. This is evident in the HBI dD record. However the abundance of this compound were relatively low, and hence we could not generate a dD record at high temporal resolution such as for C23-alkanes. This is explained in the Supplementary Information – Suitability of the chronology.

The tie-point matching presented in Figure S4 I find less convincing than the linking presented in Figure S2. If I got it right, this matching is, however, only applied in the MS to estimate uncertainties of the match presented in Figure S2, and therefore it is not as critical for the conclusions of the study.

This is correct. We only test a possible Greenland-tuned chronology to show that using the classic approach to construct marine chronologies does not significantly affect our conclusions. We have now clarified this in the caption of Supplementary Figure 5.

The ice core – C-14 synchronization presented in Figure S5 generally looks convincing. It is an approved technique for ice core (10Be) – dendrochronology (14C) synchronization and the derived offsets in chronologies of about +/-50 years agrees with those identified in other studies, such as the recent study by Adolphi et al., *Climate of the Past*, 2018. The Greenland 100 years ‘jump’ at 12.3 ka appears wired and unrealistic – if the course is a wrong tree-ring match in IntCal, it should also show up in the WD comparison? – but since all age scales show consistency within 100 years, there is no way those disagreements can generate ‘fake’ offset of time scales of about 400 years such as those identified in the present study.

We agree that the offsets between the ice core and IntCal13 time scales do not affect our conclusions. We would like to point out that the ‘jump’ at 12.5 kyr is not ‘wired’ but a real feature that has been previously observed in Muscheler et al., (2014). The ‘jump’ also appears in the comparison with WD2014 but it is more subdued because the GICC05-IntCal13 break comprises of both a tree-ring mismatch and a possible undercounting of ice layers in Greenland during GS-1 (Rasmussen et al., 2006)–which does not concern WD2014. This is now explained in the caption of Supplementary Figure 6.

In summary, I ‘buy’ the chronology part of the story, despite the somewhat controversial marine-terrestrial ‘wobble-matching’ approach. It would be convenient with another common tephra layer in the older part of the investigated time interval to verify the record alignment, but as that is currently not available, I think the MS offers the second best approach which is still convincing. I would not be surprised if later studies will be able to adjust the offsets identified in the present study, but in the deglacial period, offsets of the order of 400 yrs are too large to vanish with improved dating techniques, because the applied key chronologies are already in agreement within 100 years or better.

#####

References

- Bauch, H.A., Erlenkeuser, H., Spielhagen, R.F., Struck, U., Matthiessen, J., Thiede, J., Heinemeier, J., 2001. A multiproxy reconstruction of the evolution of deep and surface waters in the subarctic Nordic seas over the last 30,000 yr. *Quat. Sci. Rev.* 20, 659–678. [https://doi.org/10.1016/S0277-3791\(00\)00098-6](https://doi.org/10.1016/S0277-3791(00)00098-6)
- Cheng, J., Liu, Z., He, F., Otto-Bliesner, B.L., Brady, E.C., Wehrenberg, M., 2011. Simulated two-stage recovery of Atlantic meridional overturning circulation during the last deglaciation. *Abrupt Clim. Chang. Mech. Patterns, Impacts, Am. Geophys. Union* 75–92.
- Crocket, K.C., Vance, D., Gutjahr, M., Foster, G.L., Richards, D.A., 2011. Persistent Nordic deep-water overflow to the glacial North Atlantic. *Geology* 39, 515–518. <https://doi.org/10.1130/G31677.1>
- Ezat, M.M., Rasmussen, T.L., Thornalley, D.J.R., Olsen, J., Skinner, L.C., Hönisch, B., Groeneveld, J., 2017. Ventilation history of Nordic Seas overflows during the last (de)glacial period revealed by species-specific benthic foraminiferal ^{14}C dates. *Paleoceanography* 32, 172–181. <https://doi.org/10.1002/2016PA003053>
- Meland, M.Y., Dokken, T.M., Jansen, E., Hevrøy, K., 2008. Water mass properties and exchange between the Nordic seas and the northern North Atlantic during the period 23–6 ka: Benthic oxygen isotopic evidence. *Paleoceanography* 23. <https://doi.org/10.1029/2007PA001416>
- Muscheler, R., Adolphi, F., Knudsen, M.F., 2014. Assessing the differences between the IntCal and Greenland ice-core time scales for the last 14,000 years via the common cosmogenic radionuclide variations. *Quat. Sci. Rev.* 106, 81–87. <https://doi.org/10.1016/j.quascirev.2014.08.017>
- Muschitiello, F., Pausata, F.S.R., Watson, J.E., Smittenberg, R.H., Salih, A.A.M., Brooks, S.J., Whitehouse, N.J., Karlatou-Charalampopoulou, A., Wohlfarth, B., 2015. Fennoscandian freshwater control on Greenland hydroclimate shifts at the onset of the Younger Dryas. *Nat. Commun.* 6, 8939. <https://doi.org/10.1038/ncomms9939>
- Nadeau, M.-J., Grootes, P.M., Voelker, A., Bruhn, F., Duhr, A., Oriwall, A., 2001. Carbonate ^{14}C Background: Does It Have Multiple Personalities? *Radiocarbon*. <https://doi.org/10.1017/S0033822200037978>
- Rach, O., Brauer, A., Wilkes, H., Sachse, D., 2014. Delayed hydrological response to Greenland cooling at the onset of the Younger Dryas in western Europe. *Nat. Geosci.* 7, 109–112. <https://doi.org/10.1038/ngeo2053>
- Rasmussen, S.O., Andersen, K.K., Svensson, A.M., Steffensen, J.P., Vinther, B.M., Clausen, H.B., Siggaard-Andersen, M.L., Johnsen, S.J., Larsen, L.B., Dahl-Jensen, D., Bigler, M., Röthlisberger, R., Fischer, H., Goto-Azuma, K., Hansson, M.E., Ruth, U., 2006. A new Greenland ice core chronology for the last glacial termination. *J. Geophys. Res. Atmos.* 111. <https://doi.org/10.1029/2005JD006079>
- Rasmussen, T.L., Thomsen, E., 2004. The role of the North Atlantic Drift in the millennial timescale glacial climate fluctuations. *Palaeogeogr. Palaeoclimatol. Palaeoecol.*

<https://doi.org/10.1016/j.palaeo.2004.04.005>

Renold, M., Raible, C.C., Yoshimori, M., Stocker, T.F., 2009. Simulated resumption of the North Atlantic meridional overturning circulation - Slow basin-wide advection and abrupt local convection. *Quat. Sci. Rev.* 29, 101–112.
<https://doi.org/10.1016/j.quascirev.2009.11.005>

Sepulcre, S., Durand, N., Bard, E., 2017. Large ^{14}C age offsets between the fine fraction and coexisting planktonic foraminifera in shallow Caribbean sediments. *Quat. Geochronol.*
<https://doi.org/10.1016/j.quageo.2016.12.002>

Thornalley, D.J.R., Bauch, H.A., Gebbie, G., Guo, W., Ziegler, M., Bernasconi, S.M., Barker, S., Skinner, L.C., Yu, J., 2015. A warm and poorly ventilated deep Arctic Mediterranean during the last glacial period. *Science* (80-.). 349, 706–710.
<https://doi.org/10.1126/science.aaa9554>

Wohlfarth, B., Luoto, T., Muschitiello, F., Väiliranta, M., Björck, S., Davies, S.M., Kylander, M., Ljung, K., Reimer, P.J., Smittenberg, R.H., 2018. Climate and environment in southwest Sweden 15.3 – 11.3 cal. ka BP. *Boreas*.

Reviewer #1:

None

Reviewer #3:

None